# Knowledge Removal in Sampling-based Bayesian Inference

**Shaopeng Fu**[1][*]**, Fengxiang He**[2,1][*]**& Dacheng Tao**[1]
[1]The University of Sydney, Australia,    [2]JD Explore Academy, China
`shfu7008@uni.sydney.edu.au, fengxiang.f.he@gmail.com,`
`dacheng.tao@sydney.edu.au`

## Abstract

The right to be forgotten has been legislated in many countries, but its enforcement in the AI industry would cause unbearable costs. When single data deletion requests come, companies may need to delete the whole models learned with massive resources. Existing works propose methods to remove knowledge learned from data for explicitly parameterized models, which however are not appliable to the sampling-based Bayesian inference, *i.e.*, Markov chain Monte Carlo (MCMC), as MCMC can only infer implicit distributions. In this paper, we propose the first machine unlearning algorithm for MCMC. We first convert the MCMC unlearning problem into an explicit optimization problem. Based on this problem conversion, an *MCMC influence function* is designed to provably characterize the learned knowledge from data, which then delivers the MCMC unlearning algorithm. Theoretical analysis shows that MCMC unlearning would not compromise the generalizability of the MCMC models. Experiments on Gaussian mixture models and Bayesian neural networks confirm the effectiveness of the proposed algorithm. The code is available at `https://github.com/fshp971/mcmc-unlearning`.

## 1 Introduction

"The right to be forgotten" refers to the right of individuals to request data controllers such as tech giants to delete the data collected from them. It has been recognized in many countries through legislation, including the European Union's General Data Protection Regulation (2016) and the California Consumer Privacy Act (2018). In the field of machine learning, legal experts suggest that companies may need to delete and re-train the machine learning models to meet the legal requirements of the right to be forgotten (Li et al., 2018; Garg et al., 2020). However, training a machine learning model would usually cost massive amounts of resources, including data, energy, and time. Thus, the enforcement of the right to be forgotten in the AI industry may result in unbearable costs.

Toward this end, recent works (Cao & Yang, 2015; Guo et al., 2020) propose *machine unlearning algorithms* to efficiently characterize and remove the knowledge learned from specific data. To enable knowledge removal analysis, existing unlearning approaches usually require the machine learning models to be explicitly parameterized, which however do not cover an important family of machine learning algorithm, the Markov chain Monte Carlo (MCMC). MCMC is a sampling-based Bayesian inference method that aims to draw samples along a Markov chain to approximate a target posterior distribution (Hastings, 1970; Geman & Geman, 1984; Welling & Teh, 2011). Distributions inferred by MCMC are usually implicitly parameterized, which makes it difficult to directly apply existing unlearning algorithms to the MCMC unlearning problem.

In this paper, we propose an MCMC unlearning algorithm to realize the right to be forgotten in the sampling-based Bayesian inference for the first time. The key of tackling the MCMC unlearning problem is to design methods to directly analyze the implicit parameter of the learned distribution. To this end, we propose to employ the learned implicit distribution to approximate the target posterior under the measure of KL-divergence, which then converts the MCMC unlearning problem to an

---

[*]The authors contributed equally.

optimization problem with an explicit distribution parameter for knowledge removal analysis. Based on this conversion, an *MCMC influence function* is designed (Huber, 2004; Koh & Liang, 2017) to characterize the knowledge learned from single data samples. The new influence function then delivers the MCMC unlearning algorithm, which removes data knowledge by directly subtracting the corresponding influence from the learned distribution.

Theoretical analyses are conducted for the proposed MCMC unlearning algorithm. For the knowledge removal analysis, we prove that the proposed algorithm can provably realize a $\varepsilon$-knowledge removal, which is a new notation defined for evaluating the machine unlearning performance in Bayesian inference. For the generalization analysis, a PAC-Bayesian generalization upper bound (McAllester, 1998; 1999; 2003; He et al., 2019) is established for the distribution processed by the MCMC unlearning algorithm. The upper bound shows that the difference introduced by MCMC unlearning in the generalization upper bound is no larger than the order of $\mathcal{O}(\sqrt{|S'|}/N)$. This result demonstrates that the proposed algorithm would not compromise the generalization ability of the learned distribution.

In summary, our work has four main contributions: (1) We enable the knowledge removal analysis in MCMC by converting the MCMC unlearning problem to an optimization problem. (2) Based on this problem conversion, we design an MCMC influence function to characterize the knowledge learned from data, which then delivers the MCMC unlearning algorithm. (3) We conduct theoretical analyses and prove that the MCMC unlearning can realize an $\varepsilon$-knowledge removal and has little impact on the generalization ability of the learned distribution. (4) We conduct comprehensive experiments to verify the effectiveness of the proposed algorithm, which include Gaussian mixture models for clustering on synthetic data, and Bayesian neural networks for classification on real-world dataset. The results suggest that the MCMC unlearning can effectively remove the knowledge learned from the requested data while retaining the information learned from other data intact.

## 2 RELATED WORKS

**Markov chain Monte Carlo.** MCMC is a Bayesian inference method that would draw samples along a Markov chain, in which the drawn samples are proved to converge to that from a targeted distribution (Hastings, 1970). Although many improvements have been made such as Hamiltonian Monte Carlo (HMC) (Duane et al., 1987; Neal et al., 2011), Gibbs sampling (Geman & Geman, 1984; George & McCulloch, 1993) and slice sampling (Neal, 2003; Walker, 2007), MCMC still suffers from huge computational cost when inferring on large-scale dataset, *e.g.* in computer vision and image processing tasks (Zhang et al., 2020b; Wang et al., 2020), since each sampling step in MCMC requires computation over the whole dataset.

To tackle this issue, Welling & Teh (2011) introduce a scalable MCMC sampler named stochastic gradient Langevin dynamics (SGLD) that by adding a proper noise to a standard stochastic gradient optimization algorithm (Robbins & Monro, 1951), which can help the update iterations converge to samples from the true posterior distribution. Since that, a variety of stochastic gradient MCMC samplers (SG-MCMC) have been proposed, including stochastic gradient HMC (SGHMC) (Chen et al., 2014), Patterson & Teh (2013), Ahn et al. (2012), Ding et al. (2014), and Zhang et al. (2020a). Ma et al. (2015) further design a complete framework for constructing SG-MCMC.

**Machine unlearning.** Machine unlearning aims to remove the knowledge learned from specific data in efficient manners (Cao & Yang, 2015). Existing approaches can be roughly divided into *exact methods* and *approximate methods*. Exact methods aim to recover the model trained without the removed data exactly and are usually realized via reducing the retraining cost. For example, Ginart et al. (2019) design an efficient unlearning method for $k$-means clustering via model quantization. Bourtoule et al. (2021) propose a SISA framework that would only retrain the affected part of the model during data deletion. Other advances include Ullah et al. (2021) and Brophy & Lowd (2021).

On the other hand, approximate methods aim to modify the trained model to approximate that trained only on the remaining data. A series of works (Guo et al., 2020; Golatkar et al., 2020; 2021) are designed via characterizing the influence of data on the trained model with influence functions (Cook & Weisberg, 1982; Huber, 2004; Koh & Liang, 2017). Other approaches include Baumhauer et al. (2020) and Izzo et al. (2021). Some works have studied machine unlearning in Bayesian inference (Nguyen et al., 2020; Gong et al., 2021). However, these works only focus on variational

inference, an optimization-based Bayesian inference method, and could not be applied to sampling-based Bayesian inference, MCMC. In contrast, our approach is the first machine unlearning method for MCMC, with a theoretical knowledge removal guarantee.

## 3 PRELIMINARIES

Suppose $p(z|\theta)$ is a parameterized distribution over the sample space $\mathcal{Z}$, where $\theta \in \Theta \subset \mathbb{R}^d$ is the parameter with a prior distribution $p(\theta)$. Suppose $S = \{z_1, z_2, \cdots, z_N\}$ is a dataset consists of $N$ samples, where each sample $z_i \in \mathcal{Z}$ is drawn from the parameterized distribution $p(z|\theta)$, *i.e.*, $z_i \sim p(z|\theta)$. Bayesian inference aims to infer the posterior distribution $p(\theta|S)$ of the parameter $\theta$,

$$p(\theta|S) = \frac{p(\theta) \prod_{i=1}^{N} p(z_i|\theta)}{\int p(\theta) \prod_{i=1}^{N} p(z_i|\theta) \mathrm{d}\theta}.$$

For simplicity, for the given dataset $S$, we let $p_S$ denotes the true posterior distribution $p(\theta|S)$, and $\hat{p}_S$ denotes the posterior distribution that inferred by Bayesian inference.

Usually, the denominator of the posterior $p_S$ has no closed-form solution, which barriers directly calculating the posterior distribution. Thereby, researchers have designed methods to approximate the target posterior. A canonical approach is the sampling-based Bayesian inference method, Markov chain Monte Carlo (MCMC) (Hastings, 1970).

MCMC infers a posterior distribution via first constructs a Markov chain and then draws samples according to the state of the chain. When the target posterior is the stationary distribution of the Markov chain, it is proved that the drawn samples will converge to that from the target posterior. However, MCMC methods usually suffer from scalable issues on large-scale datasets.

To this end, stochastic gradient MCMC (SG-MCMC) (Ma et al., 2015) leverage mini-batch estimation (Robbins & Monro, 1951) to enable scalable sampling. In SG-MCMC, the posterior distribution $p(\theta|S)$ can be rewritten as $p(\theta|S) \propto \exp(-U(\theta))$, where $U(\theta)$ is the potential function defined as $U(\theta) = -\sum_{z_i \in S} \log p(z_i|\theta) - \log p(\theta)$. In each sampling step, drawing samples with SG-MCMC requires one to estimate the potential function $U(\theta)$ with mini-batch data $\tilde{S} \subset S$ as follows,

$$\tilde{U}(\theta) = -\frac{|S|}{|\tilde{S}|} \sum_{z_i \in \tilde{S}} \log p(z_i|\theta) - \log p(\theta).$$

Typical SG-MCMC methods include SGLD and SGHMC. Stochastic gradient Langevin dynamics (SGLD) (Welling & Teh, 2011) inserts Gaussian noise to stochastic gradients. In each sampling step, SGLD updates the posterior sample $\theta_t$ as follows,

$$\theta_{t+1} = \theta_t - \eta_t \nabla_\theta \tilde{U}(\theta_t) + 2\sqrt{\eta_t} W,$$

where $W$ is drawn from the standard Gaussian distribution $\mathcal{N}(0, I)$ and $\eta_t > 0$ is the step size.

To improve the sampling efficiency of SGLD, stochastic gradient Hamiltonian Monte Carlo (SGHMC) (Chen et al., 2014) inserts a momentum $v_t$ into the posterior sample update as below,

$$\theta_{t+1} = \theta_t + v_t,$$

$$v_{t+1} = (1 - \alpha)v_t - \eta_t \nabla_\theta \tilde{U}(\theta_t) + \sqrt{2\alpha\eta_t} W,$$

where $W$ is also drawn from the Gaussian distribution $\mathcal{N}(0, I)$, $\eta_t > 0$ is the step size, and $\alpha \in (0, 1)$ is the momentum factor. The initial momentum term $v_0$ is drawn from the Gaussian distribution $\mathcal{N}(0, \eta_0 I)$.

To ensure the drawn samples converge to the targeted posterior, the step size $\eta_t$ for both SGLD and SGHMC needs to satisfy (Welling & Teh, 2011): (1) $\sum_{t=1}^{\infty} \eta_t = \infty$; and (2) $\sum_{t=1}^{\infty} \eta_t^2 < \infty$. A typical step size schedule is $\eta_t = a(b + t)^{-r}$, where $a, b > 0$ and $r \in (0.5, 1]$.

## 4 MCMC UNLEARNING ALGORITHM

This section presents the main results of this paper. We first define a new notion, $\varepsilon$-knowledge removal, to assess the machine unlearning performance in Bayesian inference. Then, an MCMC unlearning algorithm is designed with knowledge removal and generalizability guarantees.

### 4.1 $\varepsilon$-KNOWLEDGE REMOVAL FOR BAYESIAN INFERENCE

Suppose a client requests to remove her/his data $S' \subset S$ from the whole training dataset $S$. An unlearning algorithm $\mathcal{A}$ for Bayesian inference aims to remove the knowledge learned from the requested data $S'$ from the inferred distribution $\hat{p}_S$ as follows,

$$\hat{p}_S^{-S'} = \mathcal{A}(\hat{p}_S, S'),$$

where $\hat{p}_S^{-S'}$ is named as the *processed distribution*.

In order to meet the regulation requirements, the following notion is defined to quantify the machine unlearning performance in Bayesian inference.

**Definition 1** ($\varepsilon$-knowledge removal). *For the distribution $\hat{p}_S$ that inferred on dataset $S$ via Bayesian inference and any subset $S' \subset S$, we call algorithm $\mathcal{A}$ performs $\varepsilon$-knowledge removal, if*

$$\mathrm{KL}(\hat{p}_S^{-S'} \| \hat{p}_{S-S'}) \leq \varepsilon,$$

*where $\hat{p}_S^{-S'} = \mathcal{A}(\hat{p}_S, S')$ is the processed distribution.*

**Remark 1.** *Intuitively, a smaller $\varepsilon$ indicates the algorithm $\mathcal{A}$ has better unlearning performance.*

In practice, one usually can only obtain an inferred distribution $\hat{p}_S(\cdot|\omega)$, subjects to a random seed $\omega$. In this case, the overall inferred distribution is $\hat{p}_S = \int \hat{p}_S(\cdot|\omega)p(\omega)\mathrm{d}\omega$, which suggests that the KL-divergence in Definition 1 can then be upper-bounded as below,

$$\mathrm{KL}(\hat{p}_S^{-S'} \| \hat{p}_{S-S'}) = \mathrm{KL}(\mathbb{E}_\omega \hat{p}_S^{-S'}(\cdot|\omega) \| \mathbb{E}_\omega \hat{p}_{S-S'}(\cdot|\omega)) = \int_\theta \mathbb{E}_\omega \hat{p}_S^{-S'}(\theta|\omega) \log \frac{\mathbb{E}_\omega \hat{p}_S^{-S'}(\theta|\omega)}{\mathbb{E}_\omega \hat{p}_{S-S'}(\theta|\omega)} \mathrm{d}\theta$$

$$\leq \int \mathbb{E}_\omega \left[ \hat{p}_S^{-S'}(\theta|\omega) \log \frac{\hat{p}_S^{-S'}(\theta|\omega)}{\hat{p}_{S-S'}(\theta|\omega)} \right] \mathrm{d}\theta = \mathbb{E}_\omega \mathrm{KL}(\hat{p}_S^{-S'} \| \hat{p}_{S-S'}|\omega).$$

Motivated by the above inequality, the following *knowledge removal estimator* is further defined to efficiently estimate the knowledge removal performance in practice.

**Definition 2** (knowledge removal estimator). *Suppose $\omega_1, \cdots, \omega_M$ is $M$ i.i.d. random seed in Bayesian inference. Then, the knowledge removal estimator $\hat{\varepsilon}_M$ is defined as below,*

$$\hat{\varepsilon}_M = \frac{1}{M} \sum_{m=1}^{M} \mathrm{KL}(\hat{p}_S^{-S'} \| \hat{p}_{S-S'}|\omega_m),$$

*where $\hat{p}_S^{-S'}(\cdot|\omega_m) = \mathcal{A}(\hat{p}_S(\cdot|\omega_m), S')$ is the processed distribution for the $m$-th random seed $\omega_m$.*

**Remark 2.** *Our experiments have employed this knowledge removal estimator $\hat{\varepsilon}_M$ to estimate the $\varepsilon$ value in the $\varepsilon$-knowledge removal guarantee. See Section 5.2 for details.*

**Remark 3.** *This estimator is similar to the "Local Forgetting Bound" defined in Golatkar et al. (2020). The difference is that the bound in Golatkar et al. (2020) aims to estimate the randomness introduced by some "readout functions", while $\hat{\varepsilon}_M$ aims to quantify the $\varepsilon$-knowledge removal guarantee by estimating the difference between the original and processed distributions.*

### 4.2 METHODOLOGY

This section presents the MCMC unlearning algorithm. To make the analysis easier, we assume that one can obtain the exact targeted posterior distribution via MCMC. In other words, for a posterior $\hat{p}_S$ that is inferred by MCMC, we assume that $\hat{p}_S = p_S$.

Developing unlearning algorithm for MCMC is challenging, since there is no explicitly parameterized distribution for knowledge removal analysis. To tackle this challenge, we re-formulate the MCMC unlearning problem as minimizing the following KL-divergence,

$$\min_\delta \mathrm{KL}(p_S(\cdot - \delta) \| p_{S-S'}), \tag{1}$$

where $p_S$ is the distribution already learned via MCMC, $\delta$ is the distribution shifting scale, $S'$ is the dataset that is requested to be deleted, and $p_{S-S'}$ is the targeted posterior that one aims to obtain. The

idea behinds Eq. (1) is to approximate the targeted posterior $p_{S-S'}$ via directly shifting the currently learned distribution $p_S$. In this case, one no longer needs to know the exact implicit distribution parameter to perform machine unlearning, but only the explicit shifting scale $\delta$ is enough.

So far, the MCMC unlearning problem has been converted to an optimization problem defined as Eq. (1). We then design MCMC unlearning algorithm based on this problem conversion. Let $\delta_S^{-S'}$ denotes any local minimizer of Eq. (1), and the goal of MCMC unlearning is to calculate the optimal shifting scale $\delta_S^{-S'}$. To achieve that goal, we first expand the KL-divergence in Eq. (1) as follows,

$$\mathrm{KL}(p_S(\cdot - \delta) \| p_{S-S'}) = \mathrm{KL}(p_S \| p_{S-S'}(\cdot + \delta))$$
$$= \mathbb{E}_{\theta \sim p_S}\left[\log p(\theta|S)\right] + \log p(S - S') + \mathbb{E}_{\theta \sim p_S}\left[-\log p(S, \theta + \delta) + \log p(S'|\theta + \delta)\right]. \quad (2)$$

Through Eq. (2), one can find that when removing dataset $S'$ via optimizing Eq. (1), an additional term $\mathbb{E}_{\theta \sim p_S} \log p(S'|\theta + \delta)$ is introduced to force the distribution $p_S(\cdot - \delta)$ approaching the target posterior $p_{S-S'}$. This motivate us to follow existing works (Koh & Liang, 2017; Guo et al., 2020) to conduct influence analysis on Eq. (1) based on the introduced term $\mathbb{E}_{\theta \sim p_S} \log p(S'|\theta + \delta)$.

Specifically, we define a new function $F_{-S',\tau}(\delta, S) = \mathbb{E}_{\theta \sim p_S}[-\log p(S, \theta + \delta) - \tau \cdot \log p(S'|\theta + \delta)]$, where $\tau \in [-1, 0]$. Suppose there exists a function $\hat{\delta}(\tau)$ with domain $[-1, 0]$ such that $\hat{\delta}(0) = 0$, and $\hat{\delta}(\tau)$ is a local minimizer of $F_{-S',\tau}(\delta, S)$. Then, one can let the target shifting scale $\delta_S^{-S'}$ be $\hat{\delta}(-1)$, and approximate it by the first-order approximation: $\hat{\delta}(-1) \approx \hat{\delta}(0) - \partial\hat{\delta}(0)/\partial\tau$, which means $\delta_S^{-S'} \approx -\partial\hat{\delta}(0)/\partial\tau$. Following a standard derivation, one can calculate the first derivative $\partial\hat{\delta}(0)/\partial\tau$ as the following defined *MCMC influence function* $\mathcal{I}(S')$. Thus, the optimal $\delta_S^{-S'}$ is then approximated as $\delta_S^{-S'} \approx -\mathcal{I}(S')$. For the detailed influence analysis, see Appendix A.1.

**Definition 3** (MCMC influence function). *The MCMC influence function of dataset $S' \subset S$ is defined to be*

$$\mathcal{I}(S') = -\left(\mathbb{E}_{\theta \sim p_S} \nabla_\theta^2 \log p(\theta, S)\right)^{-1} \sum_{z \in S'} \left(\mathbb{E}_{\theta \sim p_S} \nabla_\theta \log p(z|\theta)\right)^T.$$

The MCMC influence function is defined to estimate the influence of data $S'$ on the learned distribution $p_S$, as $\delta_S^{-S'} \approx -\mathcal{I}(S')$. The computational cost of the MCMC influence function would be high. An efficient calculation method is given in Appendix B.

Finally, based on Definition 3, the *MCMC unlearning algorithm* is designed as follows.

**Algorithm 1** (MCMC unlearning). *Suppose one have drawn a series of samples $\{\theta_1, \cdots, \theta_T\}$ from the posterior $p_S$ via MCMC. Then, MCMC unlearning algorithm removes the learned knowledge of dataset $S'$ from each drawn sample $\theta_i$ as follows,*

$$\theta_i' \leftarrow \theta_i - \mathcal{I}(S'),$$

*where $\mathcal{I}(S')$ is the MCMC influence function for dataset $S' \subset S$.*

**Remark 4.** *Algorithm 1 is equivalent to the following unlearning process,*

$$p_S^{-S'}(\cdot) = \mathcal{A}(p_S, S') = p_S(\cdot + \mathcal{I}(S')) \approx p_S(\cdot - \delta_S^{-S'}).$$

**Remark 5.** *Due to the inherent geometric difference between the distribution functions $p_S$ and $p_{S-S'}$, it is impossible for Algorithm 1 to directly shift $p_S$ to exactly recover $p_{S-S'}$. However, the experiments (see Section 5) show that the designed algorithm is good enough for removing learned knowledge from MCMC distributions.*

## 4.3 KNOWLEDGE REMOVAL ANALYSIS

This section studies the $\varepsilon$-knowledge removal for the proposed MCMC unlearning algorithm. We first introduces several assumptions (Assumptions 1-4) for the knowledge removal analysis.

**Assumption 1.** *Function $-\mathbb{E}_{\theta \sim p_S} \log p(\theta + \delta)$ and $-\mathbb{E}_{\theta \sim p_S} \log p(z|\theta + \delta)$ are $L_1$-Lipschitz continuous on the support set $\mathrm{supp}(\delta)$.*

The Lipschitzness of $-\mathbb{E}_{\theta \sim p_S} \log p(\theta + \delta)$ can be realized by choosing appropriate prior distribution for $\theta$, for example, let $p(\theta)$ be a Laplace distribution. Besides, the Lipschitzness of $-\mathbb{E}_{\theta \sim p_S} \log p(z|\theta + \delta)$ can usually be achieved by setting $p(z|\theta + \delta)$ to be a neural network with bounded domain in practice.

**Assumption 2.** *There exists a neighborhood $V_2 \subset \operatorname{supp}(\delta)$ of the origin point such that for any dataset $S$ and any sample point $z \in \mathcal{Z}$, the followings hold:*

- *(Smoothness). $\nabla_\delta[-\mathbb{E}_{\theta \sim p_S} \log p(\theta + \delta)]$ and $\nabla_\delta[-\mathbb{E}_{\theta \sim p_S} \log p(z|\theta + \delta)]$ are $L_2$-Lipschitz continuous on the space $V_2$.*

- *(Lipschitz Hessian). $\nabla_\delta^2[-\mathbb{E}_{\theta \sim p_S} \log p(\theta + \delta)]$ and $\nabla_\delta^2[-\mathbb{E}_{\theta \sim p_S} \log p(z|\theta + \delta)]$ are $L_3$-Lipschitz continuous on the space $V_2$.*

Assumption 2 assumes the local smoothness and Hessian Lipschitzness for the involved functions, which are common in literature for analyzing Newton methods (Nesterov & Polyak, 2006; Carmon et al., 2018; Zhou et al., 2018).

**Assumption 3** (strong convexity). *There exists a neighborhood $V_3 \subset \operatorname{supp}(\delta)$ of the origin point such that for any dataset $S$ and any sample point $z \in \mathcal{Z}$, $-\mathbb{E}_{\theta \sim p_S} \log p(\theta + \delta)$ and $-\mathbb{E}_{\theta \sim p_S} \log p(z|\theta + \delta)$ are $\mu$-strongly convex on $V_3$.*

The influence analysis and the defined MCMC influence function require the Hessian matrices of $-\mathbb{E}_{\theta \sim p_S} \log p(\theta + \delta)$ and $-\mathbb{E}_{\theta \sim p_S} \log p(z|\theta + \delta)$ to be invertible. Assumption 3 ensures the invertibilities of these Hessian matrices.

**Assumption 4.** *Suppose Assumptions 1-3 hold. Then there exists a sub-neighborhood $V \in V_2 \cap V_3$ of the origin point such that for any dataset $S$ and any real $\tau \in [-1, 0]$, the function $-\mathbb{E}_{\theta \sim p_S} \log p(\theta + \delta|S) - \tau \cdot \mathbb{E}_{\theta \sim p_S} \log p(S'|\theta + \delta)$ has at least one local minimizer falls in $V$.*

Assumption 4 assumes the target optimal solution $\delta_S^{-S'}$ of Eq. (1) exists in the subspace that we are interested in, which would make the influence analysis being meaningful.

Under Assumptions 1-4, we prove that the MCMC influence function defined in Definition 3 can rigorously estimate the influence of data on the shifting scale $\delta$.

**Theorem 1.** *Under Assumptions 1-4, we have that*

$$\delta_S^{-S'} = -\mathcal{I}(S') + \mathcal{O}\left(\frac{|S'|^2}{N^2}\right).$$

The proof of Theorem 1 is presented in Appendix A.1.

Based on Theorem 1, a knowledge removal guarantee is proved for the MCMC unlearning algorithm.

**Theorem 2.** *Suppose Assumptions 1-4 hold. Then, the MCMC unlearning algorithm $\mathcal{A}(p_S, S') = p_S(\cdot + \mathcal{I}(S'))$ performs $\varepsilon_S^{-S'}$-knowledge removal, where*

$$\varepsilon_S^{-S'} = \mathrm{KL}(p_S(\cdot - \delta_S^{-S'}) \| p_{S-S'}) + \mathcal{O}\left(\frac{|S'|^4}{N^3}\right),$$

*and $\delta_S^{-S'}$ is a local minimizer of the KL-divergence $\mathrm{KL}(p_S(\cdot - \delta) \| p_{S-S'})$.*

**Proof sketch.** We expand $\mathrm{KL}(p_S(\cdot + \mathcal{I}(S')) \| p_{S-S'})$ into Taylor series up to 2-th order around the point $\delta = \delta_S^{-S'}$. By applying the first-order optimality condition, the first-order term in the Taylor series becomes 0. By leveraging the approximation error given in Theorem 1, the second-order term is proved to be no larger than the order of $\mathcal{O}(|S'|^4/N^3)$. The proof is presented in Appendix A.2.

**Remark 6.** *As explained in the previous section, Algorithm 1 could not completely remove the learned knowledge from the inferred distribution due to the inherent geometric difference between the distribution $p_S$ and $p_{S-S'}$. Nevertheless, Theorem 2 demonstrates that the Algorithm 1 can help the KL-divergence in Eq. (1) approach its local minimum.*

## 4.4 GENERALIZATION ANALYSIS

This section studies the impact of the proposed MCMC unlearning algorithm on the generalization ability of the inferred distribution. The analysis is conducted based on the PAC-Bayes theory (McAllester, 1999; 2003; He et al., 2019), a framework for analyzing the generalization ability of stochastic machine learning models (Mohri et al., 2012; He & Tao, 2020).

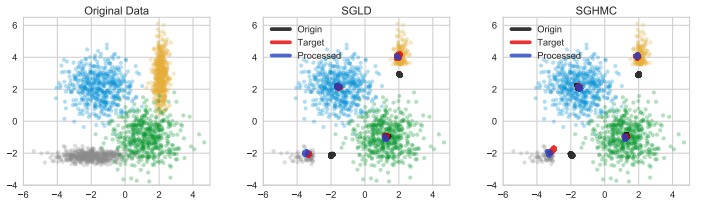

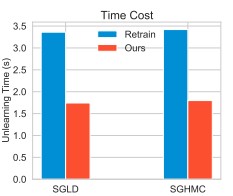

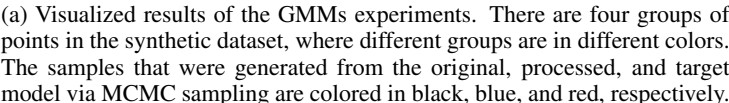

(a) Visualized results of the GMMs experiments. There are four groups of points in the synthetic dataset, where different groups are in different colors. The samples that were generated from the original, processed, and target model via MCMC sampling are colored in black, blue, and red, respectively.

(b) Time costs for processing single deletion requests via retraining and MCMC unlearning.

Figure 1: Experiment results for GMMs.

Specifically, suppose $Q$ is the parameter distribution of a stochastic model. Suppose $S$ is the training dataset. Then, the *expected risk* $\mathcal{R}(Q)$ and *empirical risk* $\hat{\mathcal{R}}_S(Q)$ of $Q$ are defined to be

$$\mathcal{R}(Q) = \underset{\theta \sim Q}{\mathbb{E}} \underset{z}{\mathbb{E}} \, \ell(\theta, z), \quad \hat{\mathcal{R}}_S(Q) = \underset{\theta \sim Q}{\mathbb{E}} \frac{1}{N} \sum_{i=1}^{N} \ell(\theta, z_i),$$

where $\theta$ is the model parameter drawn from the distribution $Q$ and $\ell$ is a loss function ranging in $[0, 1]$. The difference of the expected risk and the empirical risk is the generalization error. Its magnitude characterizes the generalization ability of the stochastic model.

Then, a generalization bound for the distribution processed by Algorithm 1 is proved as below.

**Theorem 3.** *Suppose Assumptions 1-4 hold. Let $p_S^{-S'}$ denotes the distribution processed by Algorithm 1. Then, for any $\delta \in (0, 1)$, with probability at least $1 - \delta$, the following inequality holds,*

$$\mathcal{R}(p_S^{-S'}) \leq \hat{\mathcal{R}}_S(p_S^{-S'}) + \sqrt{\frac{\mathbb{E}_{\theta \sim p_S}[\log p_S(\theta) + \|\theta\|_1] + \|\mathcal{I}(S')\|_1 + \log 2 + \log \frac{1}{\delta} + \log N + 2}{2N - 1}},$$

*where $\|\mathcal{I}(S')\|_1 \leq \mathcal{O}(|S'|/N)$.*

The proof is given in Appendix A.3.

**Corollary 1.** *Algorithm 1 increases the generalization upper bound in Theorem 3 no larger than the order of $\mathcal{O}(\sqrt{|S'|}/N)$.*

*Proof.* According to Theorem 3, the difference of the generalization upper bound introduced by Algorithm 1 is $\mathcal{O}(\sqrt{\frac{\|\mathcal{I}(S')\|_1}{2N-1}}) \leq \mathcal{O}(\sqrt{\frac{\mathcal{O}(|S'|/N)}{2N-1}}) = \mathcal{O}(\frac{\sqrt{|S'|}}{N})$. This completes the proof. $\square$

**Remark 7.** *Corollary 1 indicates that the proposed MCMC unlearning algorithm would not compromise the generalization ability of the inferred distribution.*

## 5 EXPERIMENTS

In this section, we empirically verify the effectiveness and efficiency of the proposed MCMC unlearning algorithm on the Gaussian mixture models and Bayesian neural networks.

### 5.1 EXPERIMENTS FOR GAUSSIAN MIXTURE MODELS

We first visualize the effect of the MCMC unlearning algorithm on a synthetic clustering dataset that consists of $2,000$ examples, where each examples is two-dimensional and is possibly from $4$ clusters. The Gaussian mixture models (GMMs) are employed to infer the cluster centers. For the details of the GMMs used in our experiments, see Appendix C.1.

**Experiment design.** We employ SGLD and SGHMC to infer the GMMs on the clustering dataset. Then, $400$ points are removed from each of the grey parts and yellow parts, around $40\%$ of the whole

Table 1: Experiment results on CIFAR-10. Different metrics are used to evaluate the machine unlearning performance, including the classification errors on the remaining set $S_r$, removed set $S_f$, and test set $S_{\text{test}}$, the knowledge removal estimators $\hat{\varepsilon}_M$, and the membership inference attack (MIA) accuracy on $S_f$. Every experiment is repeated 5 times. The results show that the proposed MCMC unlearning algorithm beats baseline methods in almost every experiments settings.

| | Remove | Method | Err. on $S_r$ (%) | Err. on $S_f$ (%) | Err. on $S_{\text{test}}$ (%) | $\hat{\varepsilon}_M$ ($\times 10^3$) | MIA Acc. (%) |
|---|---|---|---|---|---|---|---|
| CIFAR-10 + SGLD | 3,000 Examples | Retrain | 20.45±0.95 | 46.69±1.05 | 31.80±0.71 | 0.00±0.00 | 51.85±1.39 |
| | | Origin | 21.71±0.94 | 18.91±1.13 | 31.22±0.53 | 4079.51±2.07 | 80.11±3.87 |
| | | IS | 21.81±0.80 | 28.71±1.06 | **31.77±0.82** | 4228.11±1.60 | 72.29±2.07 |
| | | Ours | **21.56±0.96** | **31.14±1.75** | 31.76±0.66 | **4070.59±1.99** | 69.21±5.22 |
| | 5,000 Examples | Retrain | 18.61±0.90 | 100.00±0.00 | 36.25±0.63 | 0.00±0.00 | 0.00±0.00 |
| | | Origin | 21.82±0.91 | 19.07±1.26 | 31.23±0.53 | 4173.23±1.90 | 80.62±4.58 |
| | | IS | 21.17±0.87 | 54.05±1.86 | 33.25±0.77 | 4342.04±1.64 | 46.49±4.90 |
| | | Ours | **20.73±0.86** | **73.96±4.25** | **34.81±0.31** | **4151.34±1.94** | 26.64±7.68 |
| CIFAR-10 + SGHMC | 3,000 Examples | Retrain | 20.24±0.37 | 45.83±1.61 | 31.80±0.48 | 0.00±0.00 | 55.34±3.00 |
| | | Origin | 21.50±0.40 | 18.83±1.46 | 31.47±0.54 | 0.00±0.00 | 82.85±1.49 |
| | | IS | 21.61±0.40 | 28.64±1.23 | 31.90±0.53 | 4296.82±2.02 | 70.79±3.52 |
| | | Ours | **21.31±0.37** | **30.33±2.44** | **31.75±0.55** | **4137.04±2.42** | 69.83±2.78 |
| | 5,000 Examples | Retrain | 18.56±0.38 | 100.00±0.00 | 36.21±0.34 | 0.00±0.00 | 0.00±0.00 |
| | | Origin | 21.61±0.39 | 18.91±1.13 | 31.47±0.52 | 4233.95±1.73 | 82.70±1.35 |
| | | IS | 21.03±0.44 | 54.20±1.48 | 33.55±0.36 | 4408.05±1.21 | 44.20±1.63 |
| | | Ours | **20.54±0.36** | **69.56±4.45** | **34.67±0.78** | **4212.63±1.72** | 29.12±3.94 |

dataset at all, by our forgetting algorithms. We also trained models on only the remaining set with the same MCMC inference settings to show the targets of the forgetting tasks. For the details of the experiments, see Appendix C.

**Results analysis.** The visualization results are presented in Fig. 1a, which show that after unlearning, the processed models are close to the target models. Besides, the time costs for processing single data deletion requests are presented in Fig. 1b, which shows that the proposed MCMC unlearning algorithm is significantly faster than retraining the whole model. These results demonstrates that the proposed algorithm can effectively and efficiently remove the learned knowledge of specified data.

## 5.2 EXPERIMENTS FOR BAYESIAN NEURAL NETWORKS

We then apply the MCMC unlearning algorithm to the Bayesian neural networks on the CIFAR-10 (Krizhevsky et al., 2009) dataset for classification. Analogous experiments are also conducted on the Fashion-MNIST (Xiao et al., 2017) dataset, please see Appendix D for more details.

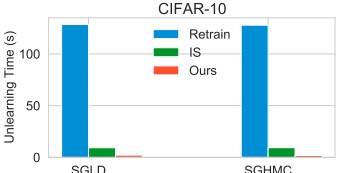

Figure 2: Time costs for processing single data deletion requests on BNNs. A smaller time cost implies a higher efficiency of the unlearning method.

**Dataset & BNN.** We divide the training set $S$ of CIFAR-10 into two parts, the removed part $S_f$ and the remained part $S_r$. A certain number of examples are randomly chosen from a single class in the dataset to form the removed training set $S_f$. The test set is denoted by $S_{\text{test}}$. A BNN consists of two convolutional layers and two fully-connected layers is adopted in the experiments. For more details about the dataset and model architecture, see Appendix D.

**Baseline method.** We compare the proposed MCMC unlearning algorithm with the importance sampling method (IS). Specifically, when a data deletion request comes, the importance sampling method will process the request via performing MCMC sampling on the remaining data.

**Evaluation metrics.** Four kinds of metrics are used to evaluate the performance of machine unlearning: (1) **Classification errors.** We calculate classification errors the remaining set $S_r$, removed set $S_f$, and test set $S_{\text{test}}$ for different models. Ideally, after removing data, the three errors of the processed model would approach that of the retrained model. (2) $\varepsilon$-**Knowledge removal guarantee.** We use the knowledge removal estimator $\hat{\varepsilon}_M$ defined in Definition 2 to estimate the $\varepsilon$ value in the $\varepsilon$-knowledge removal guarantee. A smaller estimation result would imply a better knowl-

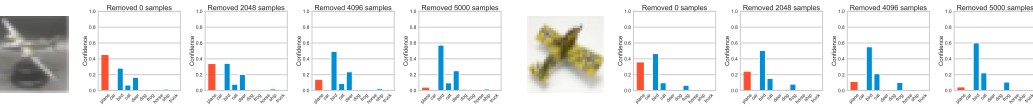

(a) The prediction confidences changes for a "plane" image. "Plane" is also the removed class.

(b) The prediction confidences changes for a "plane" image. "Plane" is also the removed class.

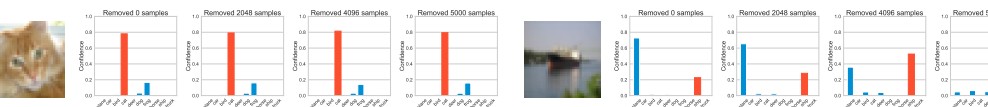

(c) The prediction confidences changes for a "cat" image. "Cat" is not the removed class.

(d) The prediction confidences changes for a "ship" image. "Ship" is not the removed class.

Figure 3: The changes of the prediction confidences for the test set examples of CIFAR-10 along with data removing. The BNN is trained with SGLD and the knowledge removal is conducted with the proposed MCMC unlearning algorithm. The predictions for the true labels are colored in red.

edge removal performance. Calculating $\hat{\varepsilon}_M$ requires to fix a series of random seeds $\omega_1, \cdots, \omega_M$. To this end, we use a series of pre-trained models as the random seeds. See Appendix D.3 for the detailed calculation procedures. (3) **Membership inference attack (MIA) accuracy.** We employ a white-box MIA (Yeom et al., 2018) to infer whether the examples from the removed subset $S_f$ come from the training set of the model. Intuitively, the attack accuracy would decline after unlearning. Thereby, a small attack accuracy implies a strong knowledge removal performance. See Appendix D.4 for more details about the MIA and its calculation procedures. (4) **Time costs for unlearning.** For different unlearning methods, we estimate the average time usages for processing single data deletion requests. A smaller time cost corresponds to high unlearning efficiency.

**Experiment designs.** We first train the BNN on the complete training set $S$ with SGLD and SGHMC, respectively. Then, the subset $S_f$ is removed iteratively, where a fixed number of data points are removed in each iteration. We also trained models on only the remaining set $S_r$ to show the targets of the machine unlearning task. For the details of the experiments, see Appendix D.5.

**Results analysis.** The experiment results are presented in Table 1. In all experiments, we observe that the proposed MCMC unlearning algorithm can significantly increase the model error on sample set $S_f$ while making the error on sample sets $S_r$ and $S_{\text{test}}$ close to that of the retrained one. Besides, our algorithm can also reduce the $\varepsilon$ value in the knowledge removal guarantee and the MIA accuracy on $S_f$, which further indicates it can indeed remove specified knowledge from the MCMC models. In contrast, the importance sampling method could neither effectively make the classification errors approach the targets, nor achieve stronger $\varepsilon$-knowledge removal. We also present the time costs for different unlearning methods in Fig. 2. One can found that the MCMC unlearning algorithm is significantly faster than other methods. All these experiment results demonstrate the effectiveness and efficiency of the proposed algorithm.

Finally, we give a case study about the prediction changes on the test set examples to illustrate the effect of the proposed MCMC unlearning algorithm. The results are presented in Fig. 3, where one can observe that as the data removing continues, the prediction confidence of the model on the removed class significantly decreases, while those on other classes remain the same or increase.

## 6 CONCLUSION

The right to be forgotten imposes a considerable compliance burden on AI companies. A company may need to delete the whole model learned from massive resources due to a request to delete a single sample point. Existing works only appliable to explicit parameterized optimization problem, which however not work for sampling-based Bayesian inference method, *i.e.*, MCMC. In this work, we convert the MCMC unlearning problem to an optimization problem. Then, an MCMC influence function is designed to characterize the influence of data on the distribution inferred by MCMC. It then delivers the first MCMC unlearning algorithm. Theoretical analyses show that the proposed algorithm can indeed reduce the learned knowledge, and would not compromise the generalization ability of the inferred distribution. Experiments show that the proposed algorithm can remove the influence of specified samples without compromising the knowledge learned on the remained data.

ACKNOWLEDGMENTS

FH is supported in part by the Major Science and Technology Innovation 2030 "New Generation Artificial Intelligence" key project (No. 2021ZD0111700). SF and DT are supported by Australian Research Council Projects FL-170100117, IH-180100002, IC-190100031, and LE-200100049. The authors sincerely appreciate Yue Xu and the anonymous ICLR reviewers for their helpful comments.

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

# A  PROOFS

This section collects all the proofs in this paper. To avoid technicalities, we assume that all functions are differentiable throughout this paper. Furthermore, the order of differentiation and integration is assumed to be interchangeable.

For simplicity, we denote that

- $h(\delta, z) := -\mathbb{E}_{\theta \sim p_S} \log p(z|\theta + \delta)$,
- $f(\delta) := -\mathbb{E}_{\theta \sim p_S} \log p(\theta + \delta)$,
- $F(\delta, S) := \sum_{z \in S} h(\delta, z) + f(\delta) = \sum_{z \in S} -\mathbb{E}_{\theta \sim p_S} \log p(z|\theta + \delta) - \mathbb{E}_{\theta \sim p_S} \log p(\theta + \delta)$.

## A.1  PROOF OF THEOREM 1

To prove Theorem 1, we first define the following function,

$$F_{-S', \tau}(\delta, S) = F(\delta, S) + \tau \sum_{z \in S'} h(\delta, z),$$

where $\tau \in [-1, 0]$.

Under Assumption 3, it can be shown that $F_{-S', \tau}(\delta, S)$ is strongly convex on the space $V_3$ defined in Assumption 3.

**Lemma 1.** *Suppose Assumption 3 holds. Then, for any $\tau \in [-1, 0]$, $F_{-S', \tau}(\delta, S)$ is strongly convex on the space $V_3$.*

*Proof of Lemma 1.* We rearrange the function $F_{-S', \tau}$ as follows,

$$F_{-S', \tau}(\delta, S) = F(\delta, S - S') + (1 + \tau) \sum_{z \in S'} h(\delta, z).$$

Apparently, $F(\delta, S - S')$ is strongly convex on $V_3$. Since $\tau \in [-1, 0]$, we have that $1 + \tau \geq 0$. Thus, $(1 + \tau) \sum_{z \in S'} h(\delta, z)$ is either strongly convex on $V_3$ or equal to zero. Therefore, $F_{-S', \tau}(\delta, S)$ is strongly convex on $V_3$.

The proof is completed. □

We then prove that when Assumption 4 holds, there exists a continuous mapping that "connects" the origin point $\mathbf{0} \in \mathbb{R}^d$ and the target shifting scale $\delta_S^{-S'}$.

**Lemma 2.** *Suppose Assumption 4 holds. Then, there exists a continuous function $\hat{\delta} : [-1, 0] \to V$ such that for any $\tau \in [-1, 0]$, $\hat{\delta}(\tau)$ is the global minimizer of the function $F_{-S', \tau}(\delta, S)$ on space $V$.*

*Proof of Lemma 2.* By applying Lemma 1 and Assumption 4, we have that the global minimizer of the function $F_{-S', \tau}(\gamma, S)$ uniquely exists in $V$. Therefore, one can define the mapping $\hat{\delta}$ as below,

$$\hat{\delta}(\tau) = \arg \min_{\delta \in V} F_{-S', \tau}(\delta, S),$$

where $\tau \in [-1, 0]$.

We then prove the continuity of $\hat{\delta}$. Notice that $\hat{\delta}(\tau)$ is the solution of the following equation,

$$\nabla_\delta F(\delta, S) + \tau \sum_{z \in S'} \nabla_\delta h(\delta, z) = 0. \tag{3}$$

Since $F_{-S', \tau}(\delta, S)$ is strongly convex, thus its Hessian matrix $\nabla_\delta^2 F_{-S', \tau}(\hat{\delta}(\tau), S)$ is invertible. Combining with the implicit function theorem, we have that $\hat{\delta}$ is continuous on the interval $[-1, 0]$.

The proof is completed. □

When $\tau$ takes 0 and $-1$, the function $\hat{\delta}(\tau)$ becomes $\mathbf{0} \in \mathbb{R}^d$ and $\delta_S^{-S'}$, respectively. Therefore, $\delta_S^{-S'}$ can be expanded into Taylor series as follows,

$$\delta_S^{-S'} = \hat{\delta}(-1) = -\frac{\partial \hat{\delta}(0)}{\partial \tau} + \frac{1}{2} \cdot \frac{\partial^2 \hat{\delta}(\xi)}{\partial \tau^2}, \tag{4}$$

where $\xi \in [-1, 0]$, $\frac{1}{2} \frac{\partial^2 \hat{\delta}(\xi)}{\partial \tau^2}$ is the Lagrange form of the remainder.

We prove that the Lagrange remainder term becomes negligible as the training set size $N$ goes to infinity. To do this, we first prove the following Lemma.

**Lemma 3.** *Suppose Assumptions 1-4 hold. The mapping $\hat{\delta}$ is as defined in Lemma 2. Then, we have that*

$$\frac{\partial \hat{\delta}(\tau)}{\partial \tau} = -\nabla_\delta^{-2} F(\hat{\delta}(\tau), S) \sum_{z \in S'} \nabla_\delta h(\hat{\delta}(\tau), z)^T,$$

*and for any $\tau \in [-1, 0]$, $\|\frac{\partial \hat{\delta}(\tau)}{\partial \tau}\|_2 \leq \mathcal{O}(|S'|/N)$.*

*Proof.* We first calculate $\frac{\partial \hat{\delta}(\tau)}{\partial \tau}$ based on Eq. (3) in the proof of Lemma 2. Calculate the derivatives of the both sides of Eq. (3) with respect to $\tau$, we have that

$$\nabla_\delta^2 F(\hat{\delta}(\tau), S) \cdot \frac{\partial \hat{\delta}(\tau)}{\partial \tau} + \sum_{z \in S'} \nabla_\delta h(\hat{\delta}(\tau), z)^T + \tau \cdot \sum_{z \in S'} \nabla_\delta^2 h(\hat{\delta}(\tau), z) \cdot \frac{\partial \hat{\delta}(\tau)}{\partial \tau} = 0. \tag{5}$$

According to Lemma 1, $F_{-S', \tau}(\delta, S)$ is strongly convex on $V$. Thus, the following Hessian matrix

$$\nabla_\delta^2 F_{-S', \tau}(\hat{\delta}(\tau), S) = \nabla_\delta^2 F(\hat{\delta}(\tau), S) + \tau \sum_{z \in S'} \nabla_\delta^2 h(\hat{\delta}(\tau), z)$$

is positive definite, hence invertible. Combining with Eq. (5), we have that

$$\frac{\partial \hat{\delta}(\tau)}{\partial \tau} = -\left( \nabla_\delta^2 F(\hat{\delta}(\tau), S) + \tau \sum_{z \in S'} \nabla_\delta^2 h(\hat{\delta}(\tau), z) \right)^{-1} \cdot \sum_{z \in S'} \nabla_\delta h(\hat{\delta}(\tau), z)^T. \tag{6}$$

We then upper bound the norm of $\frac{\partial \hat{\delta}(\tau)}{\partial \tau}$. Based on Eq. (6), we have that

$$\left\| \frac{\partial \hat{\delta}(\tau)}{\partial \tau} \right\|_2 \leq \left\| \left( \frac{1}{N} \nabla_\delta^2 F(\hat{\delta}(\tau), S) + \frac{\tau}{N} \sum_{z \in S'} \nabla_\delta^2 h(\hat{\delta}(\tau), z) \right)^{-1} \right\|_2 \cdot \left\| \frac{1}{N} \sum_{z \in S'} \nabla_\delta h(\hat{\delta}(\tau), z) \right\|_2. \tag{7}$$

We first consider the first term of the right-hand side of Eq. (7). By Assumption 3, both $f(\delta)$ and $h(\delta, z)$ are $\mu$-strongly convex on the space $V \subset \text{supp}(\delta)$. Thus, we have that

$$\frac{1}{N} \nabla_\delta^2 F(\hat{\delta}(\tau), S) + \frac{\tau}{N} \sum_{z \in S'} \nabla_\delta^2 h(\hat{\delta}(\tau), z)$$

$$= \frac{1}{N} \sum_{i=1}^N \nabla_\delta^2 h(\hat{\delta}(\tau), z_i) + \frac{1}{N} \nabla_\delta^2 f(\hat{\delta}(\tau)) + \frac{\tau}{N} \sum_{z \in S'} \nabla_\delta^2 h(\hat{\delta}(\tau), z)$$

$$\succeq \left( \frac{1}{N} \sum_{z \in S - S'} \mu + \frac{\mu}{N} + \frac{1 + \tau}{N} \sum_{z \in S'} \mu \right) I$$

$$\succeq \frac{N - |S'|}{N} \mu \cdot I. \tag{8}$$

Let $\lambda_{\min}$ denotes the smallest eigenvalue of the following matrix

$$\left( \frac{1}{N} \nabla_\delta^2 F(\hat{\delta}(\tau), S) + \frac{\tau}{N} \sum_{z \in S'} \nabla_\delta^2 h(\hat{\delta}(\tau), z) \right).$$

Then, the Eq. (8) implies that $\lambda_{\min} \geq \frac{N-|S'|}{N}\mu$. Hence, we have the following,

$$\left\| \left( \frac{1}{N}\nabla_\delta^2 F(\hat\delta(\tau), S) + \frac{\tau}{N}\sum_{z \in S'}\nabla_\delta^2 h(\hat\delta(\tau), z) \right)^{-1} \right\|_2 = \frac{1}{\lambda_{\min}} \leq \frac{N}{(N-|S'|)\mu} = \mathcal{O}(1). \quad (9)$$

We then upper bound the second term of the right-hand side of Eq. (7). By applying Assumption 1, we have that

$$\left\| \frac{1}{N}\sum_{z \in S'}\nabla_\delta h(\hat\delta(\tau), z) \right\|_2 \leq \frac{L_1 \cdot |S'|}{N} = \mathcal{O}\left( \frac{|S'|}{N} \right) \quad (10)$$

Finally, inserting eqs. (9) (10) into Eq. (7), we eventually have that

$$\left\| \frac{\partial \hat\delta(\tau)}{\partial \tau} \right\|_2 \leq \mathcal{O}(1) \cdot \mathcal{O}\left( \frac{|S'|}{N} \right) = \mathcal{O}\left( \frac{|S'|}{N} \right).$$

The proof is completed. $\qquad\square$

Then, the following Lemma is obtained based on Lemma 3, which demonstrates the strictness of the approximation in Eq. (4).

**Lemma 4.** *Suppose Assumptions 1-4 hold. The induced mapping $\hat\delta$ is as defined in Lemma 2. Then, for any $\tau \in [-1, 0]$, we have that $\|\frac{\partial^2 \hat\delta(\tau)}{\partial \tau^2}\|_2 \leq \mathcal{O}(|S'|^2/N^2)$.*

*Proof.* We first calculate $\frac{\partial^2 \delta(\tau)}{\partial \tau^2}$ based on Eq. (3). Similar to the proof of Lemma 3, we calculate the second-order derivatives of the both sides of Eq. (3) with respect to $\tau$ and have that

$$\sum_{i=1}^{N} B(\tau, z_i) + A(\tau) + \left( \sum_{i=1}^{N}\nabla_\delta^2 h(\hat\delta(\tau), z_i) + \nabla_\delta^2 f(\hat\delta(\tau)) \right) \cdot \frac{\partial^2 \hat\delta(\tau)}{\partial \tau^2}$$

$$+ 2 \cdot \sum_{z \in S'}\nabla_\delta^2 h(\hat\delta(\tau), z) \cdot \frac{\partial \hat\delta(\tau)}{\partial \tau} + \tau \sum_{z \in S'}B(\tau, z) + \tau \sum_{z \in S'}\nabla_\delta^2 h(\hat\delta(\tau), z) \cdot \frac{\partial^2 \hat\delta(\tau)}{\partial \tau^2} = 0,$$

which means

$$\frac{\partial^2 \hat\delta(\tau)}{\partial \tau^2} = - \left( \frac{1}{N}\sum_{i=1}^{N}\nabla_\delta^2 F(\hat\delta(\tau), S) + \frac{\tau}{N}\nabla_\delta^2 \sum_{z \in S'}h(\hat\delta(\tau), z) \right)^{-1}$$

$$\cdot \left( \frac{1}{N}\sum_{i=1}^{N}B(\tau, z_i) + \frac{\tau}{N}\sum_{z \in S'}B(\tau, z) + \frac{1}{N}A(\tau) + \frac{2}{N}\sum_{z \in S'}\nabla_\delta^2 h(\hat\delta(\tau), z) \cdot \frac{\partial \hat\delta(\tau)}{\partial \tau} \right), \tag{11}$$

in which the invertibility of $\left( \frac{1}{N}\sum_{i=1}^{N}\nabla_\delta^2 F(\hat\delta(\tau), S) + \frac{\tau}{N}\sum_{z \in S'}\nabla_\delta^2 h(\hat\delta(\tau), z) \right)$ is guaranteed by Eq. (8), $A(\tau), B(\tau, z) \in \mathbb{R}^{K \times 1}$, and for $i = 1, \cdots, K$, we have the following,

$$A(\tau)_i = \frac{\partial \hat\delta(\tau)}{\partial \tau}^T \cdot \nabla_\delta^2 \left( \frac{\partial f(\hat\delta(\tau))}{\partial \delta_i} \right) \cdot \frac{\partial \hat\delta(\tau)}{\partial \tau}, \tag{12}$$

$$B(\tau, z)_i = \frac{\partial \hat\delta(\tau)}{\partial \tau}^T \cdot \nabla_\delta^2 \left( \frac{\partial h(\hat\delta(\tau), z)}{\partial \delta_i} \right) \cdot \frac{\partial \hat\delta(\tau)}{\partial \tau}. \tag{13}$$

We then upper bound the norm of $\frac{\partial^2 \hat{\delta}(\tau)}{\partial \tau^2}$. Based on Eq. (11), we have that

$$\left\| \frac{\partial^2 \hat{\delta}(\tau)}{\partial \tau^2} \right\|_2$$

$$\leq \left\| \left( \frac{1}{N} \sum_{i=1}^{N} \nabla_\delta^2 F(\hat{\gamma}(\tau), S) + \frac{\tau}{N} \sum_{z \in S'} \nabla_\delta^2 h(\hat{\delta}(\tau), z) \right)^{-1} \right\|_2$$

$$\cdot \frac{1}{N} \left( \sum_{i=1}^{N} \|B(\tau, z_i)\|_2 + \tau \sum_{z \in S'} \|B(\tau, z)\|_2 + \|A(\tau)\|_2 + 2 \left\| \sum_{z \in S'} \nabla_\delta^2 h(\hat{\delta}(\tau), z) \cdot \frac{\partial \hat{\delta}(\tau)}{\partial \tau} \right\|_2 \right) \tag{14}$$

$$\leq \mathcal{O}\left( \frac{1}{N} \left( \sum_{i=1}^{N} \|B(\tau, z_i)\|_2 + \tau \sum_{z \in S'} \|B(\tau, z)\|_2 + \|A(\tau)\|_2 + 2 \left\| \sum_{z \in S'} \nabla_\delta^2 h(\hat{\delta}(\tau), z) \cdot \frac{\partial \hat{\delta}(\tau)}{\partial \tau} \right\|_2 \right) \right), \tag{15}$$

where Eq. (15) is obtained by inserting Eq. (9) (in the proof of Lemma 3) into Eq. (14). Thus, the remaining task is to upper bound the norms of $A(\tau)$, $B(\tau, z)$ and $\nabla_\delta^2 \sum_{z \in S'} h(\hat{\delta}(\tau), z) \cdot \frac{\partial \hat{\delta}(\tau)}{\partial \tau}$.

We first upper bound $\|A(\tau)\|_2$. Applying Lemma 3, we have that $\|\frac{\partial \hat{\delta}(\tau)}{\partial \tau}\|_2 \leq \mathcal{O}(|S'|/N)$. Applying the Lipschitz Hessian condition in Assumption 2, we have that $\|\nabla_\delta^2 (\frac{\partial f(\hat{\delta}(\tau))}{\partial \delta_i}\|_2$ is also bounded by a real constant. Therefore, we have that

$$\|A(\tau)\|_2 \leq \sum_{i=1}^{K} \left\| \nabla_\delta^2 \left( \frac{\partial f(\hat{\delta}(\tau))}{\partial \delta_i} \right) \right\|_2 \cdot \left\| \frac{\partial \hat{\delta}(\tau)}{\partial \tau} \right\|_2^2 \leq \sum_{i=1}^{K} \mathcal{O}(1) \cdot \mathcal{O}\left( \frac{|S'|^2}{N^2} \right) = \mathcal{O}\left( \frac{|S'|^2}{N^2} \right). \tag{16}$$

For $B(\tau, z)$, we similarly have that

$$\|B(\tau, z)\|_2 \leq \mathcal{O}\left( \frac{|S'|^2}{N^2} \right). \tag{17}$$

To upper bound the norm of $\sum_{z \in S'} \nabla_\delta^2 h(\hat{\delta}(\tau), z) \cdot \frac{\partial \hat{\delta}(\tau)}{\partial \tau}$, we apply Lemma 3, Assumption 2, and have that

$$\left\| \sum_{z \in S'} \nabla_\delta^2 h(\hat{\delta}(\tau), z) \cdot \frac{\partial \hat{\delta}(\tau)}{\partial \tau} \right\|_2 \leq \sum_{z \in S'} \left\| \nabla_\delta^2 h(\hat{\delta}(\tau), z) \right\|_2 \left\| \frac{\partial \hat{\delta}(\tau)}{\partial \tau} \right\|_2 \leq \mathcal{O}\left( \frac{|S'|^2}{N} \right). \tag{18}$$

Inserting eqs. (16), (17) and (18), into Eq. (15), we eventually have that

$$\left\| \frac{\partial^2 \hat{\delta}(\tau)}{\partial \tau^2} \right\|_2$$

$$\leq \mathcal{O}\left( \frac{1}{N} \left( \sum_{i=1}^{N} \mathcal{O}\left( \frac{|S'|^2}{N^2} \right) + \tau \cdot \mathcal{O}\left( \frac{|S'|^3}{N^2} \right) + \mathcal{O}\left( \frac{|S'|^2}{N^2} \right) + 2 \cdot \mathcal{O}\left( \frac{|S'|^2}{N} \right) \right) \right) = \mathcal{O}\left( \frac{|S'|^2}{N^2} \right).$$

The proof is completed. □

Finally, combining all the results, we prove Theorem 1.

*Proof of Theorem 1.* The proof is completed by applying Lemmas 3 and 4 into Eq. (4). □

## A.2 PROOF OF THEOREM 2

This section presents the proof of Theorem 2.

*Proof of Theorem 2.* By expanding the KL-divergence $\mathrm{KL}(p_S(\cdot + \mathcal{I}(S'))\|p_{S-S'})$ into Taylor series around the local region of $\delta = \delta_S^{-S'}$, we have

$$\mathrm{KL}(p_S(\cdot + \mathcal{I}(S'))\|p_{S-S'})$$

$$= \mathrm{KL}(p_S(\cdot - \delta_S^{-S'}\|p_{S-S'}) + (\mathcal{I}(S') + \delta_S^{-S'})^T \cdot \nabla_\delta^2 \mathrm{KL}(p_S(\cdot - \xi)\|p_{S-S'}) \cdot (\mathcal{I}(S') + \delta_S^{-S'})$$

$$= \mathrm{KL}(p_S(\cdot - \delta_S^{-S'}\|p_{S-S'}) - (\mathcal{I}(S') + \delta_S^{-S'})^T \cdot \nabla_\delta^2 \left[ F(\xi, S) - \sum_{z \in S'} h(\xi, z) \right] \cdot (\mathcal{I}(S') + \delta_S^{-S'}),$$

(19)

where the first-order term equals $0$ according to the first-order optimal condition. By applying Theorem 1 and Assumption 2, we further have

$$\left\| (\mathcal{I}(S') + \delta_S^{-S'})^T \cdot \nabla_\delta^2 \left[ F(\xi, S) - \sum_{z \in S'} h(\xi, z) \right] \cdot (\mathcal{I}(S') + \delta_S^{-S'}) \right\|_2$$

$$\leq \left\| \nabla_\delta^2 \left[ F(\xi, S) - \sum_{z \in S'} h(\xi, z) \right] \right\|_2 \cdot \left\| \mathcal{I}(S') + \delta_S^{-S'} \right\|_2^2$$

$$\leq \mathcal{O}\left( (N - |S'| + 1) L_2 \cdot \mathrm{diam}(V) \right) \cdot \mathcal{O}\left( \frac{|S'|^2}{N^2} \right)^2 \leq \mathcal{O}\left( \frac{|S'|^4}{N^3} \right). \tag{20}$$

By combining eqs. (19) and (20), the proof is completed. $\square$

## A.3 PROOF OF THEOREM 3

This section presents the proof of Theorem 3.

We derive generalization bound for the proposed algorithm under the PAC-Bayesian framework (McAllester, 1998; 1999; 2003). The framework can provide guarantees for randomized predictors (*e.g.*, the Bayesian predictors).

Specifically, let $Q$ a distribution on the parameter space $\Theta$, $P$ denotes the prior distribution over the parameter space $\Theta$. Then, the expected risks $\mathcal{R}(Q)$ is bounded in terms of the empirical risk $\hat{\mathcal{R}}(Q, S)$ and KL-divergence $\mathrm{KL}(Q\|P)$ by the following result from PAC-Bayes.

**Lemma 5** (cf. McAllester (2003), Theorem 1). *For any real $\delta \in (0, 1)$, with probability at least $1 - \delta$, we have the following inequality for all distributions $Q$:*

$$\mathcal{R}(Q) \leq \hat{\mathcal{R}}(Q, S) + \sqrt{\frac{\mathrm{KL}(Q\|P) + \log \frac{1}{\delta} + \log N + 2}{2N - 1}}. \tag{21}$$

Based on Lemma 5, we prove the generalization bounds in Theorem 3.

*proof of Theorem 3.* Let the prior distribution $P$ be a Laplace distribution $\mathrm{lap}(0, 1)$, $p_S^{-S'} = p_S(\cdot + \mathcal{I}(S'))$ be the distribution processed by the MCMC unlearning algorithm. Then, one can calculate the KL-divergence $\mathrm{KL}(p_S^{-S'}\|P)$ as follows (where we assume that $\Theta = \mathbb{R}^d$),

$$\mathrm{KL}(p_S^{-S'}\|P) = \int_\Theta \log \left( \frac{p_S^{-S'}(\theta)}{p(\theta)} \right) p_S^{-S'}(\theta) \mathrm{d}\theta$$

$$= \int_\Theta \log \left( \frac{p_S(\theta)}{p(\theta - \mathcal{I}(S'))} \right) p_S(\theta) \mathrm{d}\theta$$

$$= \int_\Theta \left[ \log p_S(\theta) + \|\theta - \mathcal{I}(S')\|_1 + \log 2 \right] p_S(\theta) \mathrm{d}\theta$$

$$\leq \mathbb{E}_{\theta \sim p_S} \log p_S(\theta) + \mathbb{E}_{\theta \sim p_S} \|\theta\|_1 + \|\mathcal{I}(S')\|_1 + \log 2. \tag{22}$$

Inserting Eq. (22) into Eq. (21) in Lemma 5, we then obtain the PAC-Bayesian generalization bound.

Eventually, by applying Lemma 3, we have that $\|\mathcal{I}(S')\|_1 \leq \mathcal{O}(|S'|/N)$.

The proof is completed. □

## B  EFFICIENT CALCULATION OF MCMC INFLUENCE FUNCTION

A major computing burden in the MCMC unlearning algorithm is calculating the product of $H^{-1}v$, where $H$ is the Hessian matrix of some vector-valued function $f(x)$ and $v$ is a constant vector. When the function $f(x)$ is of high dimension, the calculation would have a considerably high computational cost. We follow Agarwal et al. (2017) and Koh & Liang (2017) to apply a divide-and-conquer strategy to address the issue. This strategy relies on calculating the Hessian-vector product $Hv$.

**Hessian-vector product (HVP).** We first discuss how to efficiently calculate $Hv$. The calculation of $Hv$ can be decomposed into two steps: (1) calculate $\frac{\partial f(x)}{\partial x}$ and then (2) calculate $\frac{\partial}{\partial x}\left(\frac{\partial f(x)}{\partial x} \cdot v\right)$. It is worth noting that $\frac{\partial f(x)}{\partial x} \in \mathbb{R}^{1 \times d}$ and $v \in \mathbb{R}^{d \times 1}$, where $d > 0$ is the dimension of data. Thus, $\left(\frac{\partial f(x)}{\partial x} \cdot v\right)$ is a scalar value. Calculating its gradient $\frac{\partial}{\partial x}\left(\frac{\partial f(x)}{\partial x} \cdot v\right)$ has a very low computational cost on platform PyTorch (Paszke et al., 2017) or TensorFlow (Abadi et al., 2015).

**Calculating $H^{-1}v$.** When the norm $\|H\| \leq 1$, the matrix $H^{-1}$ can be expanded by the Taylor's series as $H^{-1} = \sum_{i=0}^{\infty}(I - H)^i$. Define that $H_j^{-1} = \sum_{i=0}^{j}(I - H)^i$. Then, we have the following recursive equation,

$$H_j^{-1}v = v + (I - H)H_{j-1}^{-1}v.$$

Agarwal et al. (2017) prove that when $j \to \infty$, we have $\mathbb{E}[H_j^{-1}] \to H^{-1}$. Therefore, we employ $H_j^{-1}v$ to approximate $H^{-1}v$.

Moreover, to secure the condition $\|H\| \leq 1$ stands, we scale $H$ to $cH$ by a scale $c \in \mathbb{R}^+$, such that $\|cH\| \leq 1$. Then, we approximate $(cH)^{-1}$. Eventually, we have that $H^{-1} = c(cH)^{-1}$. We can plug it to the applicable equations above.

## C  EXPERIMENTS DETAILS FOR GAUSSIAN MIXTURE MODELS

This section provides the additional experiments details for the Gaussian mixture models.

### C.1  GAUSSIAN MIXTURE MODELS

We employ Gaussian mixture models (GMMs) to infer the cluster centers on the synthetic dataset. A GMM assumes that each example is drawn from $K$ Gaussian distributions centered at $\mu_1, \cdots, \mu_K$, respectively. Then, the hierarchical structure of GMM is as follows: (1) draw a clustering center from the uniform distribution over $\{\mu_{c_1}, \ldots, \mu_{c_K}\}$; and (2) sample $z_i$ from a Gaussian distribution centering at $\mu_{c_i}$. That is,

$$\mu_k \sim \mathcal{N}(0, \sigma^2 I),$$
$$c_i \sim \text{categorical}\left(\frac{1}{K}, \cdots, \frac{1}{K}\right),$$
$$Z_i \sim \mathcal{N}(\mu_{c_i}, I),$$

where $1 \leq k \leq K, 1 \leq i \leq n, \mu_k \in \mathbb{R}^d, c_i \in \{1, \cdots, K\}, Z_i \in \mathbb{R}^d$, and the hyperparameter $\sigma \in \mathbb{R}$ is the prior standard deviation. In our experiments, the factor $K$ is set as 4, and the prior standard deviation $\sigma$ is set as 1.

### C.2  EXPERIMENTS DETAILS

The experiments for GMM have two main phases:

**Training phase.** Every GMM is trained for $4,000$ iterations. The batch size is set as $64$. For both SGLD and SGHMC, the learning rate schedule is set as $4 \cdot t^{-0.5005}/N$, where $t$ is the training iteration step and $N$ is the number of the training set size. Besides, the momentum factor $\alpha$ of SGHMC is set as $0.9$.

**Unlearning phase.** We remove a batch of $4$ datums each time. When calculating the inversed-Hessian-vector product $H^{-1}v$ in the influence functions (see Section B), the recursive calculation number $j$ is set as $32$, and the scaling factor $c$ is set as $1/N'$, where $N'$ is the number of the current remained training examples. Notice that $N'$ will gradually decrease as the forgetting process continues. Moreover, we employ the Monte Carlo method to calculate the expectations in the MCMC influence function. Specifically, we repeatedly draw sample $\theta$ for $5$ times, calculate the matrix or vector in the MCMC influence function, and average the results to approach the expectations.

## D    EXPERIMENTS DETAILS FOR BAYESIAN NEURAL NETWORKS

This section provides the additional experiments details for Bayesian neural networks.

### D.1    DATASETS

We employ two image datasets, Fashion-MNIST (Xiao et al., 2017) and CIFAR-10 (Krizhevsky et al., 2009), in our experiments. Fashion-MNIST consists of grey-scale images from $10$ classes, where each class contains $6,000$ training examples and $1,000$ test examples. Besides, CIFAR-10 consists of color images from $10$ classes, where each class contains $5,000$ training examples and $1,000$ test examples. No data augmentation is used in the experiments.

Furthermore, it is worth noting that the models used for Fashion-MNIST and CIFAR-10 are first pretrained on MNIST (LeCun et al., 1998) and CIFAR-100 (Krizhevsky et al., 2009), respectively. Please see Appendices D.3 and D.5 for more details.

### D.2    BAYESIAN NEURAL NETWORKS

This section introduces the model architectures used in our experiments and the implementation details for training BNNs.

#### D.2.1    MODEL ARCHITECTURES

For Fashion-MNIST, we use a LeNet (LeCun et al., 1998) architecture that consists of two convolutional layers followed by three fully-connected layers. The convolutional layers use $5 \times 5$ convolutions, each followed by a $2 \times 2$ max-pooling layer and a ReLU activation function. The channel numbers of the two convolutional layers are $6$ and $16$, respectively. Besides, each fully-connected layer is also followed by a ReLU activation function, while the hidden layer channels are $120$ and $84$, respectively.

for CIFAR-10, we follow Abadi et al. (2016) to use a network architecture that consists of two convolutional layers followed by two fully-connected layers. Each convolutional layer uses $5 \times 5$ convolution, with a channel number of $64$, followed by a $2 \times 2$ max-pooling layer and a ReLU activation function. Besides, each fully-connected layer is also followed by a ReLU activation function, while the hidden layer channels are $384$.

#### D.2.2    MODEL TRAINING

In all the experiments, we use an isotropic Gaussian distribution $\mathcal{N}(0, \sigma_0^2 I)$ as the prior of the BNNs, in which the hyperparameter $\sigma_0$ is set as $0.1$.

Traditional SG-MCMC methods would inject random noise to BNNs during training. However, it is found that the noise injection in the early training iterations hurts the convergence of BNNs (Zhang et al., 2020a). To alleviate such a problem, we follow Zhang et al. (2020a) to first avoid the noise injection in the early training iterations and then resume SG-MCMC as usual in the rest of the training.

### D.3 ESTIMATING THE $\varepsilon$-KNOWLEDGE REMOVAL GUARANTEE

We use the knowledge removal estimator $\hat{\varepsilon}_M$ defined in Definition 2 to estimate the $\varepsilon$ value in the $\varepsilon$-knowledge removal guarantee. According to the definition of $\hat{\varepsilon}_M$, calculating it requires first fixing a series of random seeds, and then calculating the KL-divergence between two MCMC distributions.

For the random seeds, we employ the pretraining technique (Golatkar et al., 2020) to fix them. Specifically, for the experiments on Fashion-MNIST, all the models used are first pre-trained on MNIST; for the experiments on CIFAR-10, all the models used are first pre-trained on CIFAR-100.

Besides, for the KL-divergence calculation, we follow Wang et al. (2009) to estimate the KL-divergence in a $k$-NN manner, in which 100 samples are drawn from each of the processed and retrained distributions for the estimation.

### D.4 MEMBERSHIP INFERENCE ATTACK

Membership inference attack (MIA) (Shokri et al., 2017; Yeom et al., 2018) is a privacy attack that aims to infer whether a given example is in the training set based on the output of the model. In our experiments, we adopt a white-box threshold-based MIA (Yeom et al., 2018) to assess the unlearning performance. The calculation consists of two steps: (1) learn an optimal MIA threshold on the remained training set $S_r$ and the test set $S_{\text{test}}$, and (2) calculate MIA accuracy with the learned threshold on the removed training set $S_f$.

**Learn the optimal threshold.** Suppose the "to-be-inferred" example $(x, y)$ comes from $S_r$ and $S_{\text{test}}$ with equal probabilities. Then, the attack accuracy of MIA with a threshold $\rho \in [0, 1]$ on the model $f_\theta$ is calculated as follows,

$$\text{Acc}(\rho, S_r, S_{\text{test}}) = 0.5 \times \left( \frac{\sum_{(x,y) \in S_r} \mathbf{1}[f_\theta(x)_y \geq \rho]}{|S_r|} + \frac{\sum_{(x,y) \in S_{\text{test}}} \mathbf{1}[f_\theta(x)_y < \rho]}{|S_{\text{test}}|} \right),$$

where $f_\theta(x)_y$ is the output confidence for label $y$ and $\mathbf{1}[\cdot]$ is the indicator function. Then, the optimal threshold $\rho_{\text{optim}}$ is obtained via calculating the maximizer of the above attack accuracy, *i.e.*,

$$\rho_{\text{optim}} = \arg\max_\rho \text{Acc}(\rho, S_r, S_{\text{test}}).$$

**Calculate MIA accuracy on $S_f$.** With the optimal threshold $\rho_{\text{optim}}$ that learned on $S_r$ and $S_{\text{test}}$, the MIA accuracy on the removed set $S_f$ is calculated as follows,

$$\text{Acc}(\rho_{\text{optim}}, S_f) = \frac{\sum_{(x,y) \in S_f} \mathbf{1}[f_\theta(x)_y \geq \rho_{\text{optim}}]}{|S_f|}.$$

Intuitively, after removing the knowledge learned from the removed subset $S_f$, the MIA accuracy $\text{Acc}(\rho_{\text{optim}}, S_f)$ would decline. As a result, a low MIA accuracy on the removed subset $S_f$ would imply a strong knowledge removal performance.

### D.5 EXPERIMENTS DETAILS

This section provides the omitted experiment details, including the settings of the hyperparameters and the procedures of the experiments.

#### D.5.1 EXPERIMENTS ON FASHION-MNIST

**Pretraining phase.** In every experiment, we pretrain a non-Bayesian model on MNIST via SGD for $2,000$ iterations, with a batch size of 128, a learning rate of $0.1/N$, where $N$ is the training set size, and a momentum factor of $0.9$.

**Training phase.** Every BNN is trained for $10,000$ iterations. The batch size is set as 128. For both SGLD and SGHMC, we first train the model without noise injection in the first $1,000$ iterations. In this stage, the learning rate is fixed to $0.01/N$. Then, we resume the traditional SGLD and SGHMC in the rest of the training. In this stage, the learning rate schedule is set as $0.01 \cdot t^{-0.5005}/N$, where $t$ is the training iteration step. Besides, the momentum factor $\alpha$ of SGHMC is set as $0.9$.

**Unlearning phase.** We remove a batch of $64$ datums each time.

For the MCMC unlearning algorithm, when calculating the inversed-Hessian-vector product $H^{-1}v$ in the MCMC influence function (see Appendix B), the recursive calculation number $j$ is set as $64$, the scaling factor $c$ is set as $0.05/N'$, in which $N'$ is the number of the current remained training examples. Notice that $N'$ will gradually decrease as the forgetting process continues. Besides, we employ the Monte Carlo method to calculate the expectations in the MCMC influence function. Specifically, each time we draw a sample $\theta$ via the MCMC sampler and calculate the matrix and vector in the MCMC influence function based on $\theta$ to estimate the desired expectation.

Besides, for the importance sampling method, we process each data deletion request by performing MCMC sampling for $1,000$ times on the remaining data.

### D.5.2  EXPERIMENTS ON CIFAR-10

**Pretraining phase.** In every experiment, we pretrain a non-Bayesian model on CIFAR-100 via SGD for $10,000$ iterations, with a batch size of $128$, a learning rate of $0.1/N$, where $N$ is the training set size, and a momentum factor of $0.9$.

**Training phase.** Every BNN is trained for $20,000$ iterations. The batch size is set as $128$. For both SGLD and SGHMC, we first train the model without noise injection in the first $5,000$ iterations. In this stage, the learning rate is fixed to $0.01/N$. Then, we resume the traditional SGLD and SGHMC in the rest of the training. In this stage, the learning rate schedule is set as $0.01 \cdot t^{-0.5005}/N$, where $t$ is the training iteration step. Besides, the momentum factor $\alpha$ of SGHMC is set as $0.9$.

**Unlearning phase.** We remove a batch of $64$ datums each time.

For the MCMC unlearning algorithm, When calculating the inversed-Hessian-vector product $H^{-1}v$ in the MCMC influence functions (see Appendix B), the recursive calculation number $j$ is set as $64$, the scaling factors $c$ is set as $0.08/N'$, where $N'$ is the number of the current remained training examples. Notice that $N'$ will gradually decrease as the forgetting process continues. Besides, we employ the Monte Carlo method to calculate the expectations in the MCMC influence function. Specifically, each time we draw a sample $\theta$ via the MCMC sampler and calculate the matrix and vector in the MCMC influence function based on $\theta$ to estimate the desired expectation.

Besides, for the importance sampling method, we process each data deletion request by performing MCMC sampling for $1,000$ times on the remaining data.

### D.6  EXPERIMENT RESULTS ON FASHION-MNIST

This section presents all the experiment results of BNNs on the Fashion-MNIST dataset. The evaluations of unlearning performance are shown as Table 2, while the time costs for unlearning is shown in Fig. 4 The results further justify the effectiveness of the proposed algorithm.

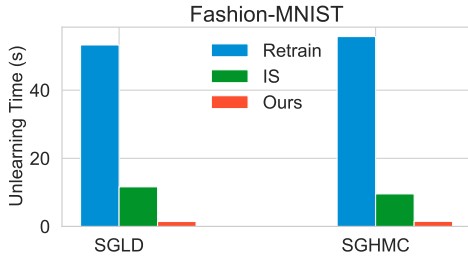

Figure 4: Time costs for processing single data deletion requests on BNNs.

### D.7  EVALUATION ON PREDICTION DIFFERENCES

In this section, we adopt the metric named *prediction differences* to better quantify the difference between the retrained and processed models. For two BNNs $p_1(\cdot)$ and $p_2(\cdot)$ learned by MCMC, their prediction difference on a given dataset $S$ is defined as $\mathbb{E}_{\theta_1 \sim p_1} \mathbb{E}_{\theta_2 \sim p_2} \frac{1}{|S|} \sum_{(x_i, y_i) \in S} \| f_{\theta_1}(x_i) -$

Table 2: Experiment results on Fashion-MNIST. Different metrics are used to evaluate the machine unlearning performance, including the classification errors on the remaining set $S_r$, removed set $S_f$, and test set $S_{\text{test}}$, the knowledge removal estimators $\hat{\varepsilon}_M$, and the membership inference attack (MIA) accuracy on $S_f$. Every experiment is repeated 5 times.

| | Remove | Method | Err. on $S_r$ (%) | Err. on $S_f$ (%) | Err. on $S_{\text{test}}$ (%) | $\hat{\varepsilon}_M$ ($\times 10^3$) | MIA Acc. (%) |
|---|---|---|---|---|---|---|---|
| Fashion MNIST + SGLD | | Retrain | 12.81±0.49 | 37.62±0.76 | 15.63±0.59 | 0.00±0.00 | 16.12±5.79 |
| | 4,000 Examples | Origin | 13.49±0.81 | 17.39±0.41 | 14.73±0.80 | 351.00±4.98 | 10.86±8.02 |
| | | IS | **12.67±0.70** | 34.93±2.16 | 15.30±0.77 | 365.97±3.04 | 13.97±11.37 |
| | | Ours | 12.93±0.72 | **35.84±1.85** | **15.57±0.77** | 350.30±4.82 | **10.36±2.31** |
| | | Retrain | 11.21±0.57 | 100.00±0.00 | 21.05±0.49 | 0.00±0.00 | 0.00±0.00 |
| | 6,000 Examples | Origin | 13.38±0.85 | 17.08±0.51 | 14.73±0.80 | 364.63±2.45 | 7.20±5.93 |
| | | IS | **11.10±0.66** | 87.30±5.82 | 19.90±0.87 | 378.11±1.45 | 4.52±2.31 |
| | | Ours | 11.39±0.68 | **98.13±1.61** | **20.59±0.99** | 362.55±2.47 | **1.25±0.67** |
| Fashion MNIST + SGHMC | | Retrain | 14.57±1.11 | 38.90±3.42 | 17.34±1.11 | 0.00±0.00 | 5.58±5.81 |
| | 4,000 Examples | Origin | 15.21±1.30 | 17.58±1.68 | 16.40±1.17 | 362.60±8.24 | 12.03±12.61 |
| | | IS | 14.27±1.22 | 36.54±1.73 | 16.97±1.16 | 375.62±5.10 | 13.76±13.12 |
| | | Ours | **14.56±1.24** | **37.48±2.02** | **17.29±1.08** | 361.72±7.87 | **11.28±13.25** |
| | | Retrain | 12.99±0.94 | 100.00±0.00 | 22.61±0.69 | 0.00±0.00 | 0.00±0.00 |
| | 6,000 Examples | Origin | 15.16±1.31 | 17.27±1.80 | 16.40±1.17 | 366.98±6.24 | 8.32±12.07 |
| | | IS | 12.52±1.04 | 87.48±7.22 | 20.94±1.44 | 382.34±3.48 | 4.20±1.84 |
| | | Ours | **12.93±1.16** | **98.23±1.04** | **22.39±0.85** | 364.99±6.00 | **1.59±0.51** |

Table 3: The results of prediction differences on CIFAR-10 and Fashion-MNIST. The prediction differences between the retrained model and the processed model are calculated on the remaining set $S_r$, removed set $S_f$, and test set $S_{\text{test}}$, respectively.

| | Remove | Method | Pred. diff. on $S_r$ | Pred. diff. on $S_f$ | Pred. diff. on $S_{\text{test}}$ |
|---|---|---|---|---|---|
| CIFAR-10 + SGLD | 3,000 Examples | Origin | 0.18±0.00 | 0.50±0.02 | 0.21±0.01 |
| | | IS | 0.18±0.00 | 0.37±0.01 | 0.20±0.01 |
| | | Ours | **0.17±0.00** | **0.35±0.02** | **0.19±0.01** |
| | 5,000 Examples | Origin | 0.23±0.00 | 1.42±0.02 | 0.37±0.01 |
| | | IS | 0.21±0.00 | 0.92±0.03 | 0.29±0.01 |
| | | Ours | **0.19±0.00** | **0.67±0.05** | **0.25±0.01** |
| CIFAR-10 + SGHMC | 3,000 Examples | Origin | 0.19±0.00 | 0.50±0.01 | 0.22±0.00 |
| | | IS | 0.19±0.00 | 0.38±0.01 | 0.21±0.00 |
| | | Ours | **0.18±0.00** | **0.35±0.01** | **0.20±0.01** |
| | 5,000 Examples | Origin | 0.23±0.00 | 1.44±0.02 | 0.37±0.00 |
| | | IS | 0.22±0.00 | 0.94±0.02 | 0.31±0.00 |
| | | Ours | **0.20±0.00** | **0.73±0.05** | **0.27±0.00** |
| Fashion MNIST + SGLD | 4,000 Examples | Origin | 0.13±0.02 | 0.37±0.02 | 0.15±0.02 |
| | | IS | **0.11±0.02** | 0.22±0.01 | **0.12±0.02** |
| | | Ours | **0.11±0.02** | **0.21±0.02** | **0.12±0.02** |
| | 6,000 Examples | Origin | 0.17±0.01 | 1.45±0.02 | 0.30±0.01 |
| | | IS | **0.13±0.01** | 0.56±0.05 | 0.18±0.01 |
| | | Ours | **0.13±0.01** | **0.33±0.07** | **0.15±0.01** |
| Fashion MNIST + SGHMC | 4,000 Examples | Origin | 0.15±0.03 | 0.43±0.03 | 0.18±0.03 |
| | | IS | **0.14±0.03** | **0.26±0.05** | **0.15±0.03** |
| | | Ours | **0.14±0.03** | **0.26±0.05** | **0.15±0.03** |
| | 6,000 Examples | Origin | 0.18±0.02 | 1.44±0.02 | 0.30±0.02 |
| | | IS | 0.14±0.02 | 0.55±0.07 | 0.18±0.02 |
| | | Ours | **0.13±0.02** | **0.35±0.05** | **0.16±0.02** |

$f_{\theta_2}(x_i)\|_1$, where $f_{\theta_1}(x_i)$ and $f_{\theta_2}(x_i)$ are two prediction confidence vectors for the example $x_i$. Intuitively, a smaller prediction difference indicates a smaller deviation between the retrained and processed models.

We calculate the prediction difference between the retrained model and the processed model on three different datasets, $S_r$, $S_f$, and $S_{\text{test}}$. The experiment results on CIFAR-10 and Fashion-MNIST are shown in Table 3. The results show that the proposed MCMC unlearning algorithm can effectively reduce the prediction difference, which further justify the effectiveness of the proposed method.

