# OpenReview forum: "Knowledge Removal in Sampling-based Bayesian Inference"
_ICLR.cc/2022/Conference — ICLR 2022 Poster_

### Official Review · Reviewer_b83d · 2021-10-31

**Correctness:** 4
**Technical Novelty And Significance:** 4
**Empirical Novelty And Significance:** Not applicable
**Recommendation:** 8
**Confidence:** 2

**Details Of Ethics Concerns:**

It is a legal requirement for companies to remove training examples requested from users. The proposed method is approximating the effect of removing those training examples, and I am not sure whether this approximated method complies with corresponding the legal requirement.

**Main Review:**

Strength:
1) The demonstrated algorithm is very simple. From the evaluation, the quality of the method also seems to be effective.
2) The proposed epsilon-knowledge removal terminology also seems to be clean. In addition, this new definition helps to evaluate the quality of the unlearning algorithm.

Weakness:
1) The paper seems to very dense on mathematical and technical details. It would be nice if authors could introduce some intuitive examples to explain how the shift is estimated.  For example, from Definition 3, the calculation of the shift requires the expectation computation under the distribution p_S. Is this estimated through samples from the p_S?
2) The evaluation of the method seems too numerical. Although I really like the metrics from the epsilon-knowledge removal terminology, some case study would make the evaluation more convincing as the problem of removing training examples is a real world problem. For example, the evaluation can be more entertaining if author can demonstrate the change on a model's prediction (both the label and confidence) on some images when similar images are removed from the training set.

Question:
Is there difficulty of extending a shift on theta to a linear transformation on theta? There seems to have some benefits of modifying each sampled modeled parameter using some scalar multiplier, as it also helps to adjust the geometry of the distribution, i.e. variance, which should be changed with the number of training examples?

**Summary Of The Paper:**

The paper demonstrates an algorithm to remove the effect of some training examples, S', on a learned model. In particular, it estimates the shift on the model parameters when the examples S' are removed, where the shift models the information contribution from the removed dataset. The shift is estimated by solving an optimization problem to minimize the KL divergence between the approximated parameter distribution using the shift and the estimating ground truth parameter distribution. Author demonstrates that the shift can be efficiently estimated using only the learning experiences that are collected with the entire dataset with theoretical approximation guarantees. Hence, one can estimate the model parameters without the knowledge of the removed the dataset and do not need to retrain the model at the same time.

**Summary Of The Review:**

The problem of removing the effect of some training examples on a trained model is well motivated by the paper.  In addition, the simplicity and the effectiveness of the proposed method should be contributory to the community.

At the same time, the reviewer is not able to verify the proofs of the theorems in 4.3 and 4.4.

---

> ### Author Response · Authors · 2021-11-19
> **To Reviewer b83d**
>
> Thank you for your constructive comments and kind support! All your concerns have been carefully addressed as below. The manuscript is carefully revised accordingly. We sincerely hope our responses fully address your questions.
>
> **Q1:** _It would be nice if authors could introduce some intuitive examples to explain how the shift is estimated. For example, from Definition 3, the calculation of the shift requires the expectation computation under the distribution p_S. Is this estimated through samples from the p_S?_
>
> **A1:** Thanks for your suggestions! We will present more intuitive examples to explain how the shift is estimated in the revised manuscript.
>
> A detailed instruction for calculating the MCMC influence function (the shift in Definition 3) is as follow; please also kindly refer to Appendix B for more details.
>
> 1. Calculate the inverse Hessian-vector-product $H^{-1} v$ ($v$ and $H$ are the gradient vector and the Hessian matrix in Definition 3) via a recursive equation,
> $$
> H_{j}^{-1} v = v + (I - H) H_{j-1}^{-1} v,
> $$
> where $I$ is the identity matrix. Please also kindly refer to [1-2] for more details.
>
> 2. Estimate the gradient vector $v$ and the Hessian matrix $H$ via samples from the p_S.
>
> So yes, the estimation is conducted through samples from the p_S.
>
> **Q2:** _The evaluation of the method seems too numerical. Although I really like the metrics from the epsilon-knowledge removal terminology, some case study would make the evaluation more convincing as the problem of removing training examples is a real world problem. For example, the evaluation can be more entertaining if the author can demonstrate the change on a model's prediction on some images when similar images are removed from the training set._
>
> **A2:** Thanks for this suggestion! We have added an additional case study to the manuscript. The case study illustrates the prediction changing process during unlearning. Please also kindly refer to Fig. 3 in the revised manuscript.
>
> **Q3:** _Is there difficulty of extending a shift on theta to a linear transformation on theta? There seems to have some benefits of modifying each sampled modeled parameter using some scalar multiplier, as it also helps to adjust the geometry of the distribution, i.e. variance, which should be changed with the number of training examples?_
>
> **A3:** Thanks for this question. Our theory suggests that the current solution has been optimal in estimating the change of the model parameter in the sense of 1st order approximation. A linear transformation on the $\theta$ may further improve the performance. However, it may have significantly large computing and memory cost, as shown in our preliminary study below. Designing an efficient low-cost algorithm for approximating the linear transformation will be a very interesting direction.
>
> Similar to Eq. (1), we consider the following optimization problem,
> $$
> \min_{W} \mathrm{KL} (p_S(\theta) \| p_{S-S’}(W \theta)),
> $$
>
> where $W \in \mathbb{R}^{d \times d}$ is the linear transformation matrix and $d$ is the dimension of $\theta$. We calculate the following influence function,
> $$
> \mathrm{vec}^{-1} \left(
> -\left(E_{\theta\sim p_S} [\nabla_{\mathrm{vec}(W)}^{2} \log p(W \theta,S)]\right)^{-1}
> \sum_{z \in S'} E_{\theta \sim p_S} [\nabla_{\mathrm{vec}(W)} \log p(z|W\theta)]
> \right),
> $$
>
> where $\mathrm{vec}(\cdot)$ is the vectorization function, $\mathrm{vec}^{-1}(\cdot)$ is the inversed vectorization function, $E_{\theta\sim p_S} [\nabla_{\mathrm{vec}(W)}^{2} \log p(W \theta,S)]$ is a Hessian matrix of dimension $(d^2 \times d^2)$, and $E_{\theta \sim p_S} [\nabla_{\mathrm{vec}(W)} \log p(z|W\theta)]$ is a gradient of dimension $d^2$.
>
> Besides, BNN is usually an over-parametrized model; $d$ would be very large. The computing and memory cost would thus be prohibitively large.
>
> [1] Agarwal N, Bullins B, Hazan E. Second-order stochastic optimization for machine learning in linear time. The Journal of Machine Learning Research. 2017.
>
> [2] Koh PW, Liang P. Understanding black-box predictions via influence functions. In International Conference on Machine Learning. 2017.

---

### Official Review · Reviewer_2pj7 · 2021-11-02

**Correctness:** 4
**Technical Novelty And Significance:** 3
**Empirical Novelty And Significance:** 2
**Recommendation:** 3
**Confidence:** 3

**Main Review:**

Strengths:

- the problem of knowledge removal seems like it may see a lot of application, and introducing the first algorithm for MCMC is an important  step

- the theoretical development of the method looks solid

Concerns / questions:

- The proposed algorithm works by translating the posterior without otherwise changing its shape. This would imply that many aspects of the data are not forgotten. (In fact, I would say that in applications where MCMC is being used to describe the shape of the posterior, then aspects of the posterior other than its location are apparently important in that application. If only the location of the posterior was important, simpler methods would have sufficed.) I think the empirical results largely hide any potential shortcomings in this area: For the simulated data, only the cluster centers are shown (Figure 1(a)), so all other aspects of the posterior remain hidden. For the BNN experiments (Table 1), none of the columns give information addressing this issue. (The column with the knowledge removal estimator has large numbers for all methods, suggesting that all result in similarly poor approximations of $\hat{p}_{S-S'}$, with the proposed method consistently marginally best. But since these are estimated upper bounds to KL-divergences, it is unclear what this implies about the actual KL-divergences of any of the methods.)

I think the inability to adjust anything about the posterior other than its location is a serious drawback. As epsilon becomes large, the concept of epsilon-knowledge removal becomes meaningless: at some point, even the original posterior (which didn't forget anything) satisfies it. The paper should be more forthcoming about these issues. Do you have a way of better quantifying how large the deviations are in both experiments?

- In section 5.2, it's unclear to me from the description what the difference is between the baseline method and just retraining.

Small remarks:

- In Table 1, for the results about CIFAR-10+SGHMC, 3000 examples, error on S_test, the score for "Ours" is bolded, but the score for IS is actually highest.

- Figure 1(b), y-axis: why does it say (/s), shouldn't that be (s)?

- Appendix C.2: "phrases" should be "phases"

**Summary Of The Paper:**

This paper proposes the first method in MCMC for the problem of knowledge removal (meaning that the sampled posterior must be modified so that some subset of the training data is "forgotten"). The paper introduces the concept of epsilon-knowledge removal to measure how closely the modified posterior corresponds to the exact posterior that would have been obtained based on only the remaining data.

**Summary Of The Review:**

This paper introduces an interesting problem, studies it well and comes up with a first algorithm. However, I have main concern about the usability of this algorithm, which only adjusts the location of the posterior while leaving its shape untouched. I am worried that under this restriction, it may be impossible to achieve the intended goal, namely forgetting part of the data. Furthermore, I think the paper should have discussed these issues with the proposed approach.

---

> ### Author Response · Authors · 2021-11-19
> **To Reviewer 2pj7 (2/2)**
>
> * **Prediction difference:** For two BNNs $p_1(\cdot)$ and $p_2(\cdot)$ learned by MCMC, their prediction difference on a given dataset $S$ is defined below,
> $$
> E_{\theta_1 \sim p_1} E_{\theta_2 \sim p_2} \left[ \frac{1}{|S|} \sum_{(x_i,y_i) \in S} || f_{\theta_1}(x_i) - f_{\theta_2}(x_i) ||_1 \right],
> $$
>
>   where $f_{\theta_1}(x_i)$ and $f_{\theta_2}(x_i)$ are two prediction confidence vectors for the example $x_i$.
>
>   We calculate the prediction difference between the retrained model and the processed model on three different datasets, $S_r$, $S_f$, and $S_{\mathrm{test}}$. Intuitively, a smaller prediction difference indicates a smaller deviation between the retrained and processed models. The results on CIFAR-10 are shown in the following Tables 2-3.
>
>   | Method | Pre. on $D_r$ | Pre. on $D_f$ | Pre. on $D_{\mathrm{test}}$ |
> | ------ | ------------- | ------------- | --------------------------- |
> | Origin | 0.41          | 1.51          | 0.58                        |
> | IS     | 0.38          | 1.1           | 0.52                        |
> | Ours   | **0.37**      | **0.9**       | **0.49**                    |
>
>   *Table 2: Prediction difference for "CIFAR-10 + SGLD + Remove 5000 examples".*
>
>   | Method | Pre. on $D_r$ | Pre. on $D_f$ | Pre. on $D_{\mathrm{test}}$ |
> | ------ | ------------- | ------------- | --------------------------- |
> | Origin | 0.4           | 1.51          | 0.58                        |
> | IS     | 0.38          | 1.1           | 0.51                        |
> | Ours   | **0.37**      | **0.94**      | **0.49**                    |
>
>   *Table 3: Prediction difference for "CIFAR-10 + SGHMC + Remove 5000 examples".*
>
>   The tables show that our method can effectively unlearn the knowledge learned from the requested data and reduce the deviation between original and retrained models, which fully supports the claims of our paper. Please kindly refer to Appendix D.7 in the revised manuscript for more details.
>
> **Q1.2:** _I think the inability to adjust anything about the posterior other than its location is a serious drawback. As epsilon becomes large, the concept of epsilon-knowledge removal becomes meaningless: at some point, even the original posterior (which didn't forget anything) satisfies it. The paper should be more forthcoming about these issues. Do you have a way of better quantifying how large the deviations are in both experiments?_
>
> **A1.2:** Thanks. We agree that $\varepsilon$ cannot be large; $\varepsilon$ will be set as a considerably small value in practice to secure the unlearning performance.
>
> Besides, we have adopted a new metric, **prediction differences**, to better quantify the deviation between MCMC models. Please see **A1.1** for more details.
>
> **Q2:** _In section 5.2, it's unclear to me from the description what the difference is between the baseline method and just retraining._
>
> **A2:** The baseline method fine-tunes the original model learned on the remained data (obtained by removing requested data from the full data), while just retraining method retrains a model from scratch on the remained data.
>
> **Q3:** _In Table 1, for the results about CIFAR-10+SGHMC, 3000 examples, error on S_test, the score for "Ours" is bolded, but the score for IS is actually highest._
>
> **A3:** We respectfully note this score (error on $S_{\mathrm{test}}$) is “the closer to ‘Retrain’ the better” (which is exactly our case) but “the higher the better”. In some ways, the goal of machine unlearning is to modify the trained model to approximate the retrained model.
>
> **Q4:** _Figure 1(b), y-axis: why does it say (/s), shouldn't that be (s)?_
>
> **A4:** Thanks and addressed.
>
> **Q5:** _Appendix C.2: "phrases" should be "phases"._
>
> **A5:** Thanks and addressed.

---

> ### Author Response · Authors · 2021-11-19
> **To Reviewer 2pj7 (1/2)**
>
> Thank you for your thorough review and constructive comments. All your concerns have been carefully addressed as below. The manuscript is carefully revised accordingly. We sincerely hope our responses fully address your questions.
>
> **Q1.1:** _The proposed algorithm works by translating the posterior without otherwise changing its shape. This would imply that many aspects of the data are not forgotten. (In fact, I would say that in applications where MCMC is being used to describe the shape of the posterior, then aspects of the posterior other than its location are apparently important in that application. If only the location of the posterior was important, simpler methods would have sufficed.) I think the empirical results largely hide any potential shortcomings in this area: for the simulated data, only the cluster centers are shown (Figure 1(a)), so all other aspects of the posterior remain hidden; for the BNN experiments (Table 1), none of the columns give information addressing this issue. (The column with the knowledge removal estimator has large numbers for all methods, suggesting that all result in similarly poor approximations of $\hat p_{S-S’}$, with the proposed method consistently marginally best. But since these are estimated upper bounds to KL-divergences, it is unclear what this implies about the actual KL-divergences of any of the methods.)_
>
> **A1.1:** Thanks for your insightful question. We have conducted additional experiments according to your suggestions, which fully support our claims.
>
> **Simulation:** we first train models on the full dataset. This model is processed by our proposed method. We then retrain this model on the dataset where the requested data is removed. These three models are termed as original model, processed model, and retrained model for the brevity. The models are then employed to generate samples, which are visualized in Fig. 1(a) in the revised manuscript. The visualization shows that the samples generated by the processed model has a very closed shape with those generated by the retrained model.
>
> **BNN:** we  first train models on the full dataset. This model is processed by our proposed method. We then retrain this model on the dataset where the requested data is removed. These three models are termed as original model, processed model, and retrained model for the brevity. Two new empirical metrics are then introduced, the membership inference attack accuracy (MIA Acc.) on the requested data, and the prediction difference between the retrained and the unlearned models, in the rebuttal session. The empirical results for the new metrics are presented below.
>
> * **MIA Acc:** BNN is first trained on CIFAR-10. Then, our algorithm is applied to unlearn the knowledge learned from the requested data. A BNN is also retrained on the remained data where the requested data is removed. These three models are termed as original model, processed model, and retrained model for the brevity. Membership inference attack is applied to the processed model to measure how likely the requested data appears in the training data. Intuitively, a larger membership inference attack accuracy indicates a worse unlearning performance. The results on CIFAR-10 are shown in the following Table 1.
>
>   | Experiment Type              | Origin     | IS         | Ours           |
> | ---------------------------- | ---------- | ---------- | -------------- |
> | SGLD + Remove 5000 examples  | 80.62±4.58 | 46.49±4.9  | **26.64±7.68** |
> | SGHMC + Remove 5000 examples | 82.7±1.35  | 44.2±1.63  | **29.12±3.94** |
>
>   *Table 1: Results of Membership inference attack accuracy on the Removed subset $D_f$ of CIFAR-10. A smaller attack accuracy would imply a strong knowledge removal performance.*
>
>   The table shows that our method can effectively unlearn the knowledge learned from the requested data, which fully supports the claims of our paper. Please kindly refer to Section 5.2
> in the revised manuscript for more details.

---

### Official Review · Reviewer_VQTF · 2021-11-03

**Correctness:** 4
**Technical Novelty And Significance:** 3
**Empirical Novelty And Significance:** 3
**Recommendation:** 8
**Confidence:** 2

**Main Review:**

Strong Points

- This work addresses a relevant problem that has not been addressed before in the literature. The  “the right to be forgotten” is part of the legislation of many countries. AI-Industry needs tools to implement this right. Previous literature do not cover the case where the machine learning model is represented as a sequence of samples (i.e. Markov Chain Monte Carlo)

- The proposed method is sound and provides theoretical guarantees about the capacity of the presented approach to implement this task. This is something that is really needed in this context.

- A well-performed experimental evaluation shows the effectiveness of the presented method.


Weak Points

- The theoretical guarantees provided in this work are based on several technical assumptions which are hard to verify in real contexts. The question is: Is this proposed method an acceptable solution in a legal context? Your approach provides theoretical guarantees that it will erase this information from the model (with an epsilon error), but this guarantee only applies if a set of assumptions are met. But it is almost impossible to verify that theses assumptions are met in a real context. In consequence, will the legislator be willing to accept this method as a solution to implement “the right to be forgotten”? This is not discussed in the paper.


**Summary Of The Paper:**

This paper approaches the problem of how to make a machine learning model forgets some data samples it was trained on. This is of high relevance in the context of “the right to be forgotten” legislation. In this case, this work approaches this problem for machine learning models represented as Markov Chain Monte Carlo, which is of relevance in the context of Bayesian learning. The presented approach is based on the formalization of the problem as an optimization problem. Authors also provide theoretical guarantees that the presented approach really makes the Markov Chain Monte Carlo forgets a given set of samples.

**Summary Of The Review:**

I lean to accept this paper. I think they present a solid contribution to a relevant problem. Even though, there are some limitations that should be discussed by the authors.

---

> ### Author Response · Authors · 2021-11-19
> **To Reviewer VQTF (2/2)**
>
> **Prediction difference:** For two BNNs $p_1(\cdot)$ and $p_2(\cdot)$ learned by MCMC, their prediction difference on a given dataset $S$ is defined below,
> $$
> E_{\theta_1 \sim p_1} E_{\theta_2 \sim p_2} \left[ \frac{1}{|S|} \sum_{(x_i,y_i) \in S} || f_{\theta_1}(x_i) - f_{\theta_2}(x_i) ||_1 \right],
> $$
>
> where $f_{\theta_1}(x_i)$ and $f_{\theta_2}(x_i)$ are two prediction confidence vectors for the example $x_i$.
>
> We calculate the prediction difference between the retrained model and the processed model on three different datasets, $S_r$, $S_f$, and $S_{\mathrm{test}}$. Intuitively, a smaller prediction difference indicates a smaller deviation between the retrained and processed models. The results on CIFAR-10 are shown in the following Tables 2-3.
>
> | Method | Pre. on $D_r$ | Pre. on $D_f$ | Pre. on $D_{\mathrm{test}}$ |
> | ------ | ------------- | ------------- | --------------------------- |
> | Origin | 0.41          | 1.51          | 0.58                        |
> | IS     | 0.38          | 1.1           | 0.52                        |
> | Ours   | **0.37**      | **0.9**       | **0.49**                    |
>
> *Table 2: Prediction difference for "CIFAR-10 + SGLD + Remove 5000 examples".*
>
> | Method | Pre. on $D_r$ | Pre. on $D_f$ | Pre. on $D_{\mathrm{test}}$ |
> | ------ | ------------- | ------------- | --------------------------- |
> | Origin | 0.4           | 1.51          | 0.58                        |
> | IS     | 0.38          | 1.1           | 0.51                        |
> | Ours   | **0.37**      | **0.94**      | **0.49**                    |
>
> *Table 3: Prediction difference for "CIFAR-10 + SGHMC + Remove 5000 examples".*
>
> The tables show that our method can effectively unlearn the knowledge learned from the requested data and reduce the deviation between original and retrained models, which fully supports the claims of our paper. Please kindly refer to Appendix D.7 in the revised manuscript for more details.
>
> [1] Li H, Xu Z, Taylor G, Studer C, Goldstein T. Visualizing the Loss Landscape of Neural Nets. Advances in Neural Information Processing Systems. 2018.
>
> [2] Santurkar S, Tsipras D, Ilyas A, Madry A. How Does Batch Normalization Help Optimization? Advances in Neural Information Processing Systems. 2018.

---

> > ### Comment · Reviewer_VQTF · 2021-12-02
> > **I vote for acceptance**
> >
> > Dear authors,
> >
> > thanks for your response. I think your comments mainly address my concerns. Please, include them in the final version of the paper.

---

> > > ### Author Response · Authors · 2021-12-02
> > > **Thanks!**
> > >
> > > Dear Reviewer VQTF,
> > >
> > > Thank you very much for your support! All of them will be included in our paper.
> > >
> > > Best regards,
> > > The authors

---

> ### Author Response · Authors · 2021-11-19
> **To Reviewer VQTF (1/2)**
>
> Thank you for your constructive comments and kind support! All your concerns have been carefully addressed as below. The manuscript is carefully revised accordingly. We sincerely hope our responses fully address your questions.
>
> **Q1:** _The theoretical guarantees provided in this work are based on several technical assumptions which are hard to verify in real contexts. The question is: Is this proposed method an acceptable solution in a legal context? Your approach provides theoretical guarantees that it will erase this information from the model (with an epsilon error), but this guarantee only applies if a set of assumptions are met. But it is almost impossible to verify that these assumptions are met in a real context. In consequence, will the legislator be willing to accept this method as a solution to implement “the right to be forgotten”? This is not discussed in the paper._
>
> **A1:** Thanks for this interesting question!
>
> We agree that implementing “the right to be forgotten” is a complicated multidiscipline task. It needs efforts from both AI and legal aspects.
>
> It is a very interesting future direction to study our proposed method from the legal context. It would significantly prompt the legislation and further enforcement of the right to be forgotten. We will closely collaborate with legal experts on this topic in the future.
>
> In this paper, we mainly focus on the aspect of AI. We provide a low-cost technical solution for realizing the right to be forgotten. We verify this solution by theoretical analysis and empirical evaluation. We agree that the theory relies on several assumptions. However, most of the assumptions are testable and verifiable by numerical experiments, and are met in most existing practical learning algorithms. Assumptions 1-3 respectively assume that $-E_{\theta \sim p_S} \log p(\theta+\delta)$ (the difference between the learned posterior and the prior) and $-E_{\theta \sim p_S} \log p(z|\theta+\delta)$ (the knowledge learned from single data) are *locally* Lipschitz continuous, continuously 1st order differentiable, and continuously 2nd order differentiable, in *the neighborhood of the estimated probability*. Moreover, Assumption 4 intuitively assumes that the learned model parameter is continuous with respect to the training data. These assumptions rigorously stand in many conventional methods, such as GMM. In deep learning, these assumptions coincide with existing visualizations that the neighborhood of the estimated probability can be very smooth and convex, when some normal techniques are used, such as residual connections, weight decay, and batch normalization [1, 2].
>
> Further, we have designed quantitative metrics for evaluating the forgetting performance, including the test error on the requested data and the factor $\epsilon$ in the $\epsilon$-knowledge removal guarantee. We have introduced two new empirical metrics, the membership inference attack accuracy (MIA Acc.) on the requested data, and the prediction difference between the retrained and unlearned models, in the rebuttal session. Legislators may set thresholds of these metrics for qualified learning algorithms.
> The empirical results of the new metrics are presented below.
>
> **MIA Acc:** BNN is first trained on CIFAR-10. Then, our algorithm is applied to unlearn the knowledge learned from the requested data. A BNN is also retrained on the remained data where the requested data is removed. These three models are termed as original model, processed model, and retrained model for the brevity. Membership inference attack is applied to the processed model to measure how likely the requested data appears in the training data. Intuitively, a larger membership inference attack accuracy indicates a worse unlearning performance. The results on CIFAR-10 are shown in the following Table 1.
>
> | Experiment Type              | Origin     | IS         | Ours           |
> | ---------------------------- | ---------- | ---------- | -------------- |
> | SGLD + Remove 5000 examples  | 80.62±4.58 | 46.49±4.9  | **26.64±7.68** |
> | SGHMC + Remove 5000 examples | 82.7±1.35  | 44.2±1.63  | **29.12±3.94** |
>
> *Table 1: Results of Membership inference attack accuracy on the Removed subset $D_f$ of CIFAR-10. A smaller attack accuracy would imply a strong knowledge removal performance.*
>
> The table shows that our method can effectively unlearn the knowledge learned from the requested data, which fully supports the claims of our paper. Please kindly refer to Section 5.2
> in the revised manuscript for more details.

---

### Official Review · Reviewer_uE21 · 2021-11-03

**Correctness:** 4
**Technical Novelty And Significance:** 4
**Empirical Novelty And Significance:** 4
**Recommendation:** 8
**Confidence:** 3

**Main Review:**

2. Rationale for the score

The problem is very nicely introduced and well justified with recent (GDPR!) developments in application of IT / AI System.

The contribution is well structured into four main parts, each part being then outlined in the paper.

The supplementary material is complete and provides a nice benefit.

The Choice of BNNs and GMMs as use cases for MCMC learned distributions is well made, both algorithms being widely used and rather important scenarios for such MCMC distributions.

The choice of real world dateets (FMNIST and CIFAR-10 ) is taking well known but complex enough datasets into account.

The paper is well structured, graphical contributions in the main part as well as in the appendix serve to enable a good understanding of the approach done.

Weaknesses: None


**Summary Of The Paper:**

1. Summary

The paper proposes a novel method for knowledge removal in a MCMC context. The contribution is four-fold: First, a theoretical analysis relates the MCMC unlearning problem to an optimiziation problem introduced. Second, the conversed problem is related to a MCMC influence function. Third an abduction of theoretical results is done. Finally real world data experiments are performed showing that the proposed approach can indeed tackle the problem introduced.

**Summary Of The Review:**

A very nice paper, both well justified and nicely outlined. I can't see any weaknesses.

---

> ### Author Response · Authors · 2021-11-18
> **To Reviewer uE21**
>
> Thank you very much for your thorough review and kind support!

---

### Decision · Program_Chairs · 2022-01-20

**Decision:**

Accept (Poster)

**Comment:**

The article proposes an approach to alter the approximate posterior distribution in order to remove some of the information (unlearning). The approach is applicable when the approximate posterior is obtained via (stochastic gradient) MCMC methods. Unlearning is done by shifting the approximate true posterior by some value delta, which is found via optimisation with an influence function.

The approach is novel, tackles an important problem and is mathematically sound. Reviewers have highlighted some of its limitations. In particular, the modified posterior is obtained by translating the original posterior, without changing its shape; many aspects of the data may therefore not be forgotten. The authors have partially addressed this concern in their response and provided additional experiments. Although there is still disagreement amongst reviewers, I recommend acceptance.

A Minor comment:
I would not present MCMC merely as a "machine learning algorithm" (p.1), nor as a "sampling based Bayesian inference method" (p.1 and p.2). MCMC is a generic approach to approximate high-dimensional integrals and obtain samples approximately sampled from some target distribution, dating back from the work of Metropolis (1953) and Hastings (1970). Its application to Bayesian inference/machine learning problems came much later, see e.g. the excellent review of C. Robert and R. Casella: A short History of MCMC: Subjective recollections from incomplete data. Statistical Science, 2011.